Subject Area:
genetics/cellular biology/molecular biology

Keywords:
mitochondrial DNA, heteroplasmy, purifying selection, selfish selection, mitochondrial replacement therapy

Author for correspondence:
Hansong Ma
e-mail: hm555@cam.ac.uk

# A battle for transmission: the cooperative and selfish animal mitochondrial genomes

Anna Klucnika[1,2] and Hansong Ma[1,2]

[1]Wellcome Trust/Cancer Research UK Gurdon Institute, Tennis Court Road, Cambridge CB2 1QN, UK
[2]Department of Genetics, University of Cambridge, Downing Street, Cambridge CB2 3EH, UK

AK, 0000-0001-6102-7141; HM, 0000-0002-2705-1970

The mitochondrial genome is an evolutionarily persistent and cooperative component of metazoan cells that contributes to energy production and many other cellular processes. Despite sharing the same host as the nuclear genome, the multi-copy mitochondrial DNA (mtDNA) follows very different rules of replication and transmission, which translate into differences in the patterns of selection. On one hand, mtDNA is dependent on the host for its transmission, so selections would favour genomes that boost organismal fitness. On the other hand, genetic heterogeneity within an individual allows different mitochondrial genomes to compete for transmission. This intra-organismal competition could select for the best replicator, which does not necessarily give the fittest organisms, resulting in mito-nuclear conflict. In this review, we discuss the recent advances in our understanding of the mechanisms and opposing forces governing mtDNA transmission and selection in bilaterians, and what the implications of these are for mtDNA evolution and mitochondrial replacement therapy.

## 1. Background

Mitochondria, the powerhouse of the cell, have attracted increasing attention because of their fascinating biology and health connections. They are thought to have evolved from free-living bacteria via symbiosis, which changed the course of eukaryotic evolution through a monumental metabolic upgrade by employing oxygen to produce energy [1,2]. While now tightly integrated into the biology of the host cell, with most proteins encoded in the nuclear genome, mitochondria still retain a reduced but vital genome of their own known as mitochondrial DNA (mtDNA). The genetic content and organization of mtDNA can vary incredibly among different species (summarized in [3,4]). For bilaterians, which are the focus of this review, mtDNA is often a compact circular DNA molecule with no introns and very few intergenic regions. It usually encodes 13 proteins of the respiratory chain complex, two ribosomal RNAs (rRNAs) and 22 transfer RNAs (tRNAs). The genome also contains a distinct non-coding region/control region that encompasses replication origin(s) and transcription promoters (figure 1).

Unlike the nuclear genome, which represents an assorted mixture of both maternal and paternal DNA, animal mtDNA is normally inherited exclusively from the mother. As such, the maternal genomes do not face any heredity competitors from the male parent and can safely assume their places in the next generation. Yet, not all maternal genomes are the same [9]. As most cells contain hundreds or even thousands of copies of mtDNA, spontaneous and inherited mutations can occur in a subpopulation, creating heteroplasmic organisms with genetic diversity in the mtDNA population. Theoretically, constantly occurring mutations would make heteroplasmy a default state. Even if the selection is actively removing mutant genomes, a return to homoplasmy can

royalsocietypublishing.org/journal/rsob    Open Biol. 9: 180267

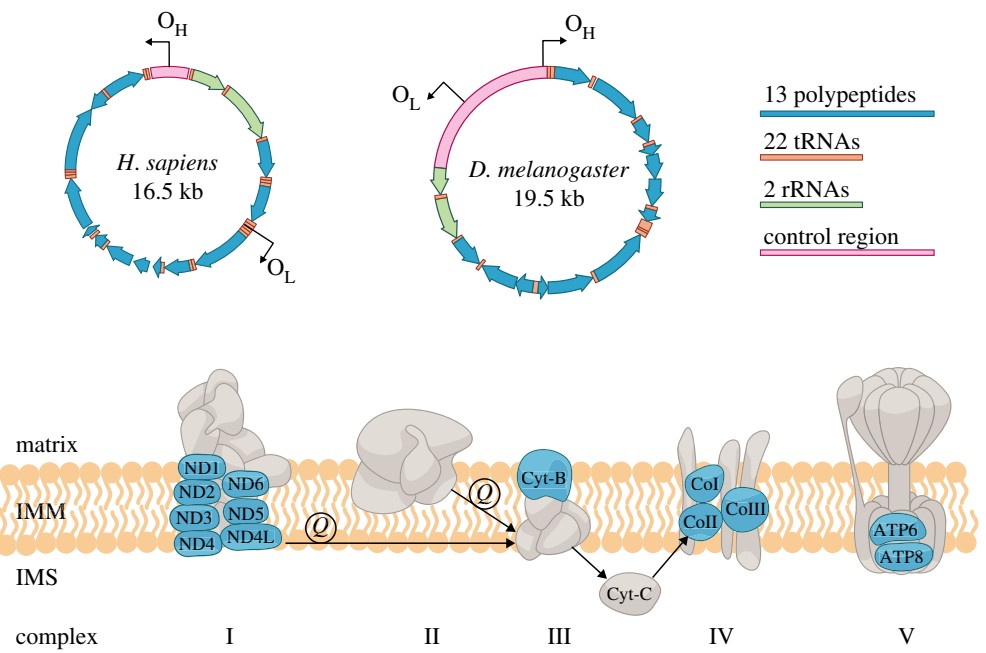

**Figure 1.** Map of the human (*Homo sapiens*) and *Drosophila melanogaster* mtDNA, representative of the mammalian and insect genome, respectively. Both genomes have the same coding capacity, but differ in gene order, length of the control region and location of the replication origins ($O_L$, light chain; $O_H$, heavy chain). The 13 polypeptides (blue) form the respiratory chain complex together with the nuclear-encoded proteins (grey) [5]. In addition, a small peptide named humanin is encoded in the 16S rRNA gene of the human mtDNA. Humanin has been shown to have a role in regulating stress resistance and conferring specific protection against Alzheimer's disease [6–8]. IMM, inner mitochondrial membrane; IMS, intermembrane space; Q, the ubiquinone form of CoQ10.

take time, resulting in transient heteroplasmy. Indeed, modern high-throughput sequencing provides evidence of widespread low-level heteroplasmy in many tissues of healthy individuals in humans [10–13]. Extensive heteroplasmy has also been reported in a number of other species including rabbits, horses, macaques, ferrets, cats and dogs [14–16]. In rare cases, heteroplasmy can be created by paternal leakage in animals that follow strict maternal inheritance [17–23]. In over 100 species of different bivalve orders, heteroplasmy occurs in male somatic tissues owing to doubly uniparental inheritance, where the female genome is transmitted to both male and female soma, and also female gonad, while the male genome is transmitted only to the male soma and gonad [24]. Among bilaterians, doubly uniparental inheritance is very much an exception to the rule with probably a single evolutionary origin [25].

Heteroplasmy can represent a dynamic and constantly changing mtDNA population within an organism [26] (figure 2). This is because individual mtDNA molecules do not replicate in equal numbers in dividing cells, nor do they turn over at equal rates in non-dividing cells. By chance, a variant molecule may replicate more frequently than the wild-type genome and thus increase in abundance. mtDNA also lacks segregation mechanisms that ensure unbiased transmission into daughter cells, so the genome can be under the strong influence of genetic drift [27–29]. Besides random fluctuation, selection can further change heteroplasmy levels; mitochondrial genomes that provide better respiratory function might be preferentially transmitted owing to positive or purifying selection, while genomes that have a replicative advantage will increase in abundance through selfish selection (i.e. selection for selfish gains in transmission). Moreover, germline bottlenecks [30–35] and occasional recombination [36–43] can quickly shift mtDNA from one subpopulation to another within individuals and between generations.

When the abundance of pathogenic mutations reaches a threshold level, physiological consequences will become apparent (reviewed by [44,45]). To date, over 350 pathogenic mitochondrial mutations have been reported to cause a spectrum of mitochondrial diseases [46], for which there are still no cures. One emerging strategy to prevent the transmission of mitochondrial mutations to offspring is mitochondrial replacement therapy (MRT), which has been approved in the UK as part of *in vitro* fertilization (IVF) treatment since 2015 [47]. MRT involves the transfer of the nucleus from a fertilized or unfertilized egg which carries mitochondrial mutations into an enucleated egg of a healthy donor, producing 'three-parent babies'. However, carryover of pathogenic mtDNA has been observed in multiple experimental trials using human or rhesus macaque eggs [48–54], and also in the first child born from MRT [55]. Even though the carried over mutants often account for less than 2% of total mtDNA, they may increase in abundance in somatic and germline tissues of those born from MRT as the individuals develop and age, and cause mitochondrial diseases later in life or in their children.

Heteroplasmy creates a battlefield for coexisting mitochondrial genomes to compete for transmission. There could be conflicts between the cooperative interest enforced by the nuclear genome and the selfish interest of the mitochondrial genome. The outcome of the competition has profound and incompletely understood impacts on the accumulation of mtDNA mutations during development and ageing, the progression and phenotypic complexity of mitochondrial disease, the inheritance of mitochondrial mutations from mother to progeny and the effectiveness of MRT. This review focuses on some of the recent efforts to investigate how different types of selection shape bilaterian mtDNA evolution within individuals and between generations, and how unexpected interactions can compromise the efficacy of MRT.

royalsocietypublishing.org/journal/rsob    Open Biol. 9: 180267

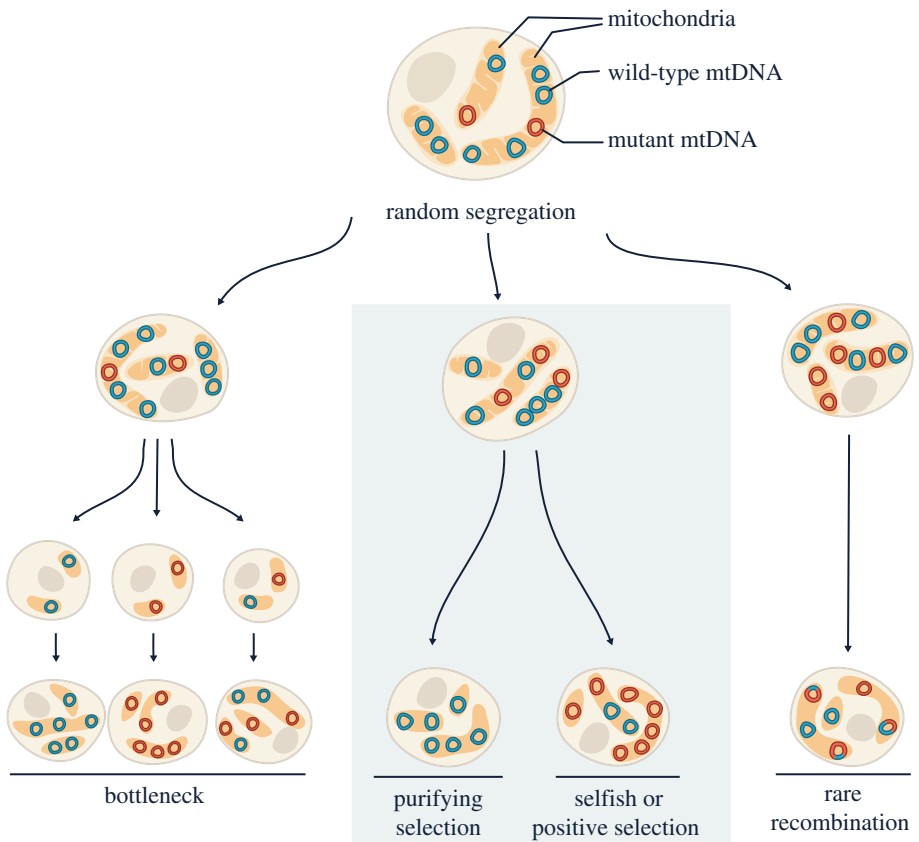

**Figure 2.** Heteroplasmy dynamics during somatic and germline transmission of mtDNA. In each cell, mitochondrial genomes are dispersed throughout the dynamic mitochondrial network and are packed in nucleoid structures, with each nucleoid containing one or more copies of mtDNA. As the cell divides, relaxed replication and random segregation of mtDNA create daughter cells with different heteroplasmy levels, while often maintaining total mtDNA copy number. The shift in the heteroplasmy level can be accelerated when there is a sharp decline in the number of transmitted mtDNA (i.e. genetic bottleneck, left panel). Besides neutral drift, selections can further alter heteroplasmy levels in a biased manner (middle panel). Very occasionally, recombination events can create hybrid genomes and alter the heteroplasmy composition (right panel).

## 2. Selections for organismal fitness

All gene products of mtDNA are devoted to energy production via oxidative phosphorylation (OXPHOS), which is of paramount importance to the host. However, the mitochondrial genome is vulnerable to mutational meltdown because uniparental inheritance and little recombination has limited power of removing *de novo* mutations. A small proportion of these mutations have been shown to be adaptive and have experienced positive selection. For instance, high-altitude populations in Tibet show adaptive mtDNA haplotypes compared with low-altitude, related groups in humans [56,57], grasshoppers [58] and horses [59,60] (reviewed by Luo *et al.* [61]). Similarly, mtDNA haplotypes have been shown to be positively selected in populations owing to their effect on tolerance to local temperatures in humans [62] and in other animals [63–67]. However, a larger proportion of mitochondrial mutations are deleterious, and purifying selection is known to be the dominant force to purge these mutations and keep the functional integrity of mitochondrial genes.

The presence of purifying selection is reflected by the fact that mitochondrial-encoded proteins evolve much more slowly than predicted [68]. In addition, several multi-generational experiments in mouse and *Drosophila* have shown that purifying selection in the female germline reduced the transmission of detrimental mtDNA mutations [69–73] (also recently reviewed by [26,74–76]). In humans, two studies

which analysed heteroplasmy transmission in mother–child pairs of European ancestry using blood or buccal mtDNA data showed a significant decrease in minor non-synonymous alleles in offspring mtDNA [35,77]. More recently, sequencing mtDNA of human primordial germline cells (PGCs) isolated from various embryonic stages revealed a reduced number of non-synonymous and tRNA mutations during PGC development [78]. Although pre-existing differences in the heteroplasmy level of different tissues or embryos could contribute to the observed decline in mutation load, the above studies suggest that purifying selection is likely to occur in the female germline in humans.

Intra-organismal purifying selection could act at the level of the cell, organelle or genome (figure 3). In some organisms, mitochondrial genetic bottlenecks (figure 2) in the germline facilitate selection at the cell level: cells that inherit more mutant mtDNA are less fit, so are less likely to propagate further. In zebrafish [79], sheep [80], mice [31,32] and humans [78], there is a dramatic decline in mtDNA copy number in PGCs. In *Caenorhabditis elegans*, PGCs form lobes that are removed and digested by endodermal cells, dramatically reducing the total amount of mitochondria in those cells [81]. In humans, oocytes were found to contain an average of $1.22 \times 10^6$ copies of mtDNA, while PGCs contained just 1425 copies on average, with an estimated five copies per mitochondrion [78]. This reduction in mtDNA copy number during germline development has been proposed to cause large shifts in heteroplasmy level between generations [30–32,35]. In addition, a

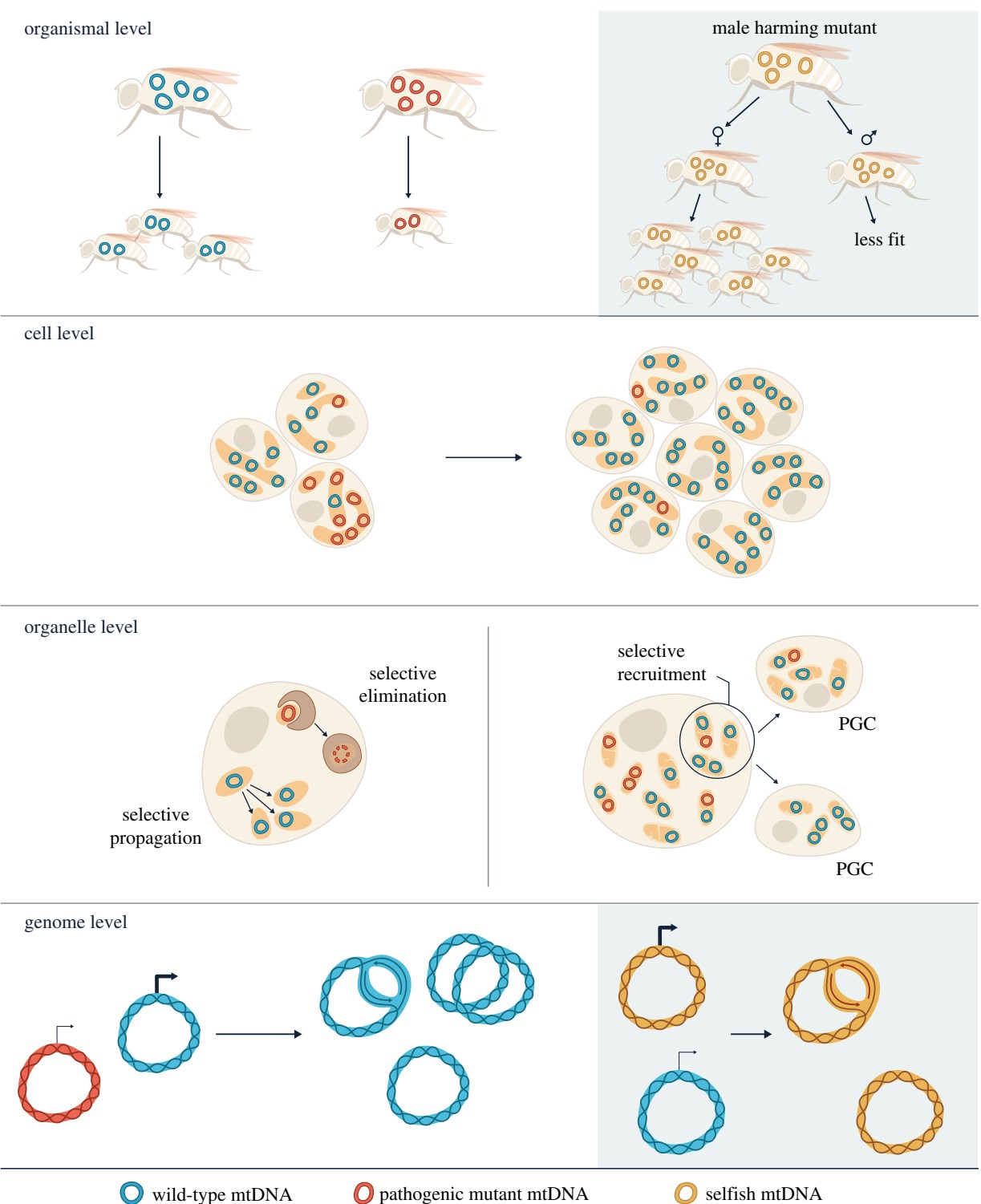

**Figure 3.** Selective transmission of mtDNA can be achieved by multiple mechanisms operating at both organismal and intra-organismal levels. Purifying selection (clear background) can occur at the organismal level owing to reduced host fitness caused by accumulation of mtDNA mutations. This mode of selection is more effective when the mutation level is high because the coexisting functional genomes can mask the physiological effects of low-abundance mutants. Within individuals, purifying selection can occur through selective propagation of more functional cells, mitochondria and mtDNA in germline and soma. For selfish gains in transmission (shaded background), mutations that are male harming, but neutral or beneficial to female fitness, can increase their abundance. This is because maternal inheritance limits the scope of purifying selection at the organismal level against such mutations among the male population. Within individuals, selfish transmission is often due to gains in replication.

bottleneck may result from unequal segregation or replication of mitochondrial genomes [33,34]. The bottleneck may even occur post-fertilization when there is rapid zygotic division unaccompanied by mtDNA replication, producing somatic cells with different mitochondrial mutation loads (e.g. [82]). Such bottlenecks may increase inter-cellular variation of the mtDNA pool, and thus accelerate purifying selection at the cell level.

Purifying selection may occur at the level of the organelle (figure 3), although it is still unclear how the OXPHOS function of individual mitochondria or mitochondrial networks is sensed to achieve selection. For example, selective recruitment or active propagation of functional mitochondria to the germplasm has been suggested by studies in zebrafish and *Drosophila* [83,84]. In *Drosophila*, there is also evidence

linking mtDNA replication to OXPHOS function, such that purifying selection occurs by preferential replication of functional mtDNA [73]. The $mt:CoI^{ts}$ genome is a temperature-sensitive lethal mutant isolated through a selection method based on expressing a restriction enzyme targeted to the mitochondria in the female germline [85]. The defect is due to a missense mutation in the coding region of cytochrome $c$ oxidase I ($CoI$). When heteroplasmic flies containing $mt:CoI^{ts}$ and wild-type mtDNA were created by cytoplasmic transfer, the level of $mt:CoI^{ts}$ decreased over generations at the restrictive temperature, and this eventually led to its elimination [72,73]. Hill *et al.* [73] showed that $mt:CoI^{ts}$ underwent reduced replication in early oogenesis and that reduced mtDNA replication also occurred when mitochondrial function was impaired by other means, such as uncoupling drugs. In order for this mechanism of selection to be effective, mtDNA must be relatively homoplasmic within an organelle when selection occurs. Indeed, increased fission of mitochondria before mtDNA replication was observed, suggesting a low mtDNA copy number per mitochondrion [73]. It is still unclear how preferential mtDNA replication is achieved during oogenesis or whether there are other mechanisms of selection acting in the germline simultaneously (e.g. [86–88]), especially since selective elimination of $mt:CoI^{ts}$ was not observed in post-development somatic tissues in the same heteroplasmic flies [89].

As an alternative mechanism of selection at the organelle level, mitochondria with mutant mtDNA can be eliminated by mitochondrial quality control mechanisms such as mitophagy (reviewed by Pickles *et al.* [90]). In a study using *D. melanogaster* heteroplasmic for both wild-type and a deletion-bearing mtDNA variant in their post-mitotic flight muscle, overexpression of some autophagy and mitophagy proteins (e.g. Atg8a, PINK1 and Parkin) promoted selective removal of deletion-bearing molecules [91]. In addition, decreased mitofusin levels, which limit the ability of mitochondrial fragments to re-fuse with the network, enhanced the removal of deletion-bearing mtDNA in the flight muscle [91]. However, the selective elimination of $mt:CoI^{ts}$ in the fly germline, as described earlier, did not seem to require Parkin [72]. Furthermore, knockout of Parkin did not affect the level of somatic mtDNA mutations in mutator mice, which are known to accumulate mtDNA mutations, although it did lead to more mitochondrial dysfunction in those mice [92]. These studies show that Parkin-mediated mitophagy may not always play a role in eliminating mtDNA mutations and reveals the diverse nature of purifying selection.

While there is ample evidence in favour of purifying selection, there are also examples where purifying selection was not detected. Many population data in humans find that segregation of mutations appears to follow random genetic drift without selection [93–96]. In a mouse model containing two mutant genomes among the wild-type genome, one mutant mtDNA that contained a missense mutation in *ND6* was rapidly eliminated within a few generations, whereas the other mtDNA containing a missense mutation in *CoI* that causes myopathy and cardiopathy persisted [69]. Similarly, Freyer *et al.* [71] showed that mitochondrial mutations in protein-coding genes were preferentially eliminated in mice over generations, whereas mutations in tRNA genes evaded selection, despite the fact that many of these mutations are potentially pathogenic.

Therefore, whether purifying selection takes place or not seems to depend on the nature of the mutation, the competing mitochondrial genomes, the tissue and the nuclear background. It is most likely that the term 'purifying selection' summarizes a plethora of selective phenomena that could differ completely for the underlying mechanisms, resulting in the complex dynamics of heteroplasmy observed.

# 3. Selections for selfish mtDNA

In addition to favouring traits that enhance organismal fitness, evolution favours selfish traits that give replication or transmission gains. Both mitochondrial and nuclear genomes are selected to maximally propagate the genes comprising its own set, independently of the effect on the other gene set or host. With few constraints on replication and segregation of mtDNA, free-wheeling intra-organismal competition is likely to select for the best replicator, regardless of its OXPHOS output.

The occurrence of selfish transmission is hard to detect in natural populations, as its consequence only becomes obvious when the selfish genome also possesses a detrimental mutation. Even if a detrimental selfish mtDNA does arise, its increase in abundance can run the risk of lowering host fitness to the point where it drives the host, and therefore itself, to extinction. Nevertheless, male harming mtDNA variants that are neutral or beneficial to female fitness can reach high frequencies in populations because males are a dead-end for mtDNA transmission (figure 3). This is known as the mother's curse and has been primarily studied in plants (summarized in [97]). A number of cases have also been recently reported in *Drosophila* [98,99] and in a human population in Canada [100].

Selfish mutations that reduce both male and female fitness are less common, but have been found in natural populations of *Drosophila subobscura* [101], *C. elegans* [102] and *Caenorhabditis briggsae* [103]. In all cases, the selfish genomes that exhibited long-term persistence in isolated strains were mtDNA variants with a large deletion. For the *D. subobscura* strain, the mutant genome contains a 5 kb deletion affecting 10 genes and accounts for approximately 80% of the total mtDNA copies [101]. The stable transmission of the deletion molecule is unlikely owing to physical attachment to the wild-type mtDNA because both genomes are autonomous monomers [104]. In *C. elegans*, the *uaDf5* mitochondrial genome, which has a 3.1 kb deletion that removes 11 genes, accounts for approximately 60% of the total mtDNA copies in heteroplasmic animals [102]. *uaDf5* was shown to be stably transmitted for over 100 generations, during which not a single line homoplasmic for wild-type or *uaDf5* mtDNA arose [102]. In another nematode species, *C. briggsae*, many natural lineages are heteroplasmic for mtDNA with a 786 bp deletion in the *ND5* coding region (*nad5Δ*) [103]. This deletion mutant is found in several geographical locations, indicating its evolutionary persistence. Clark *et al.* [105] investigated the inheritance patterns of *nad5Δ* in eight lineages and found a uniform bias towards the inheritance of a greater proportion of the *nad5Δ* genome, despite that high levels of *nad5Δ* caused reduced fertility and pharyngeal pumping rates. It is currently unclear how these deletion-bearing molecules persist in wild populations. A recent study suggested that *uaDf5* can somehow run away from the copy number control mechanism

royalsocietypublishing.org/journal/rsob Open Biol. 9: 180267

that regulates the coexisting wild-type mtDNA levels because they observed a wider variation in *uaDf5* copy number relative to that of wild-type [106]. An increase in the total mtDNA copy number could be an attempt to alleviate OXPHOS deficiency as proposed by previous theoretical work [106–109]. Furthermore, high levels of *uaDf5* activated the mitochondrial unfolded protein response, which was suggested to help the maintenance of the *uaDf5* genome [106,110].

More evidence of selfish transmission has recently emerged from experimentally generated heteroplasmic lines in *D. melanogaster*, where diverged mitochondrial genomes from different *D. melanogaster* strains or even different *Drosophila* species were paired for competition [111]. Apparently, these diverged genomes often do not compete based on their OXPHOS function. In one example, the temperature-sensitive mutant *mt:CoI^ts^* displaced a complementing genome, leading to population death after several generations at the restrictive temperature. As mentioned earlier, the *mt:CoI^ts^* mutant genome is eliminated by purifying selection when paired with a closely related wild-type *D. melanogaster* mtDNA [72,73]. However, when it was paired with another functional but more diverged genome called *ATP6[1]*, which is a *D. melanogaster* mtDNA variant that differs from *mt:CoI^ts^* by multiple single nucleotide polymorphisms (SNPs) and indels in both coding and non-coding regions, the level of *mt:CoI^ts^* increased from around 20% to 90% within four generations. Eventually, the *ATP6[1]* genome declined to the extent that it could no longer sustain the life of the flies [111] (figure 4*a*). In this case, the *mt:CoI^ts^* mutant was considered to have a selfish drive. Interestingly, while most lines ended in lethality, a few survived [42]. In these lines, recombinant mtDNA were generated which combined the functional *CoI* allele from the *ATP6[1]* genome with the selfish drive from *mt:CoI^ts^*. Once emerged, such recombinant genomes quickly outcompeted coexisting *mt:CoI^ts^* because of purifying selection and the stock became healthier over time. Since all recombinant genomes retained the non-coding region of *mt:CoI^ts^*, which differs significantly from the *ATP6[1]* genome at the sequence level, this region is believed to be responsible for the strong selfish drive. That is why the selfish drive of *mt:CoI^ts^* is not apparent when paired with its closely related wild-type *D. melanogaster* mtDNA as they share the same non-coding region. The non-coding region contains the origins of replication, so the selfish drive in this case has been linked to replicative advantage. It is worth noting that the non-coding region of *mt:CoI^ts^* is significantly longer than that of the *ATP6[1]* genome. This is surprising as the mitochondrial genomes with the smallest and least redundant DNA are believed to go to fixation within cells, organisms and then populations [112–114]. This example implies that other factors besides genome size can play a more important role in selfish transmission.

Another example of selfish mtDNA was revealed by a number of cross-species pairings in the same study. Ma and O'Farrell [111] created *D. melanogaster* flies with only wild-type *D. yakuba* mtDNA via cytoplasmic transfer followed by expression of a mitochondrially targeted restriction enzyme that will only cut the *D. melanogaster* mtDNA [111]. Despite that *D. melanogaster* and *D. yakuba* diverged about 10 million years ago, the *D. melanogaster* (mito-*D. yakuba*) flies are as healthy as wild-type, indicating that *D. yakuba* mtDNA can provide the wild-type level of function in the *D. melanogaster* nuclear background. Nevertheless, when *D. yakuba* mtDNA

was placed in competition with a number of functionally compromised *D. melanogaster* mtDNA variants, it was quickly outcompeted despite providing better function. In this case, the *D. melanogaster* mtDNA variants had a selfish advantage over *D. yakuba* mtDNA. Interestingly, mtDNA from *Drosophila mauritiana* (a species diverged ∼2 million years ago) can outcompete endogenous *D. melanogaster* mtDNA with no deleterious effect, indicating that the home genome is not always the winner [111,115]. De Stordeur [116] also used cytoplasmic transfer to study competition between different *Drosophila simulans* mtDNA haplotypes and found a hierarchy of which haplotypes could overtake which others, although it is not clear whether selfish selection plays a role in each context. Overall, these examples suggest that the mismatches in competitive strength are common among diverged genomes.

Of note, selfish selection can be neutral to the host when the selfish drive is not linked to detrimental mutations. It can even be beneficial if a more functional genome gains a replicative advantage, as it will speed up the takeover of the functional genome. For instance, Rand [117] has shown that spontaneous mutations that increase the length of the non-coding region of *Drosophila* mtDNA could occur in natural populations. These long variants were preferentially transmitted to the offspring, but there was no evidence for fitness difference among flies carrying mtDNA variants of different length [117]. In such cases, selfish selection results in rapid divergence of mtDNA sequence among different female lineages. This is because, during evolution, constant waves of taking over by a new mutant genome with replicative advantage will continuously select for the best replicator in a given nuclear background, especially if that mutation does not result in functional sacrifices. Uniparental inheritance limits mitochondrial variants to evolve in individual lineages, so, within each lineage, different winning mutations are likely to be fixed independently. As the non-coding region is often linked to selfish drive, selfish selection can accelerate divergence of this region. Indeed, for most mitochondrial genomes sequenced so far, their non-coding regions are highly variable [111].

## 4. The interplay of different types of selection at multiple levels

The outcome of mtDNA competition will depend on the relative strength of purifying and selfish selection. These two forces can oppose one another at both organismal and intra-organismal levels. In cases where deleterious mutations are linked to a strong selfish drive, they will quickly accumulate within individuals. This will eventually lower the fitness of the host and trigger purifying selection at the organismal level. In this way, a selfish element drives its own extinction. When such an element arises *de novo*, maternal inheritance restricts it to a single female lineage, and thus facilitates its elimination without spreading the detrimental effect to the rest of the population [118]. Diverse mechanisms have been described that eliminate paternal mtDNA before and/or after fertilization in different species to ensure maternal inheritance [119–125] (summarized by Sato & Sato [126]). Nevertheless, paternal leakage has been reported in multiple cases [17–23], and it is unclear to what extent rare leakage can affect the spread of selfish mtDNA within a species.

royalsocietypublishing.org/journal/rsob   Open Biol. 9: 180267

royalsocietypublishing.org/journal/rsob   Open Biol. **9**: 180267

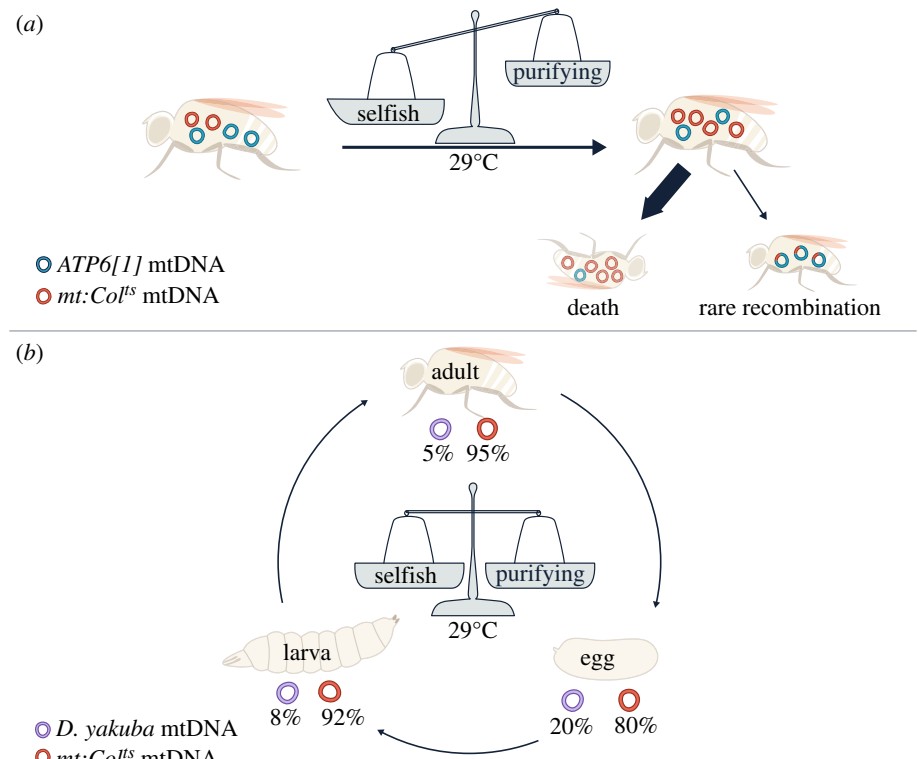

**Figure 4.** Different selective pressures dynamically act on mtDNA in *D. melanogaster* [111]. (*a*) When the *mt:Col^ts* mutant was paired with a diverged, functional *ATP6[1]* genome at the restrictive temperature, selfish selection dominated and allowed the *mt:Col^ts* genome to take over, despite purifying selection against *mt:Col^ts*. This led to the death of the population within a few generations. However, in three out of the 51 heteroplasmic lineages, recombination generated mtDNA containing the functional coding region of *ATP6[1]* and the selfish non-coding region of *mt:Col^ts* [42]. Once emerged, these recombinants became dominant owing to purifying selection against *mt:Col^ts*, allowing the flies to survive the selection. (*b*) When *mt:Col^ts* was paired with wild-type *Drosophila yakuba* mtDNA at the restrictive temperature, the relative proportion of each genome remained stable over many generations. Interestingly, the heteroplasmy level differed at various developmental stages. This is likely to be due to a dynamic balance of purifying and selfish selection in different tissues and at different developmental stages.

Within individuals, opposing types of selection may explain why sometimes purifying selection was not detected or was not always complete, resulting in the persistence of detrimental variants [69,76]. This may occur if the selfish drive only gives the linked detrimental allele a small gain in replication or transmission, which can be counterbalanced by purifying selection at the cell or organelle level. Some of the long-term heteroplasmy discussed earlier could also be due to balanced purifying and selfish selection occurring within individuals. For instance, in *C. elegans*, the *uaDf5* genome appears to be under opposing selective pressures which have different strengths at different levels of heteroplasmy [102]. When the proportion of *uaDf5* in hermaphrodites is high, the average *uaDf5* content in progeny is significantly lower. Conversely, when the proportion of *uaDf5* in hermaphrodites is low, the average *uaDf5* contents in progeny increases significantly. These data suggest the existence of at least two opposing forces, with one force leading to the increased proportion of *uaDf5* mtDNA when its levels are low, while the second force leads to decreased proportions of *uaDf5* mtDNA when its levels are high [102]. In *Drosophila*, opposing selfish and purifying selection was shown to counterbalance in a cross-species heteroplasmic line, allowing stable transmission of the functional *D. yakuba* mtDNA at 5% and the selfish detrimental *D. melanogaster mt:Col^ts* mutant at 95% in adults at the restrictive temperature [111]. How the two types of selection counteract can be complex. Selfish selection mainly operates at the genome level, whereas purifying selection can occur at genome, organelle and cell level. Furthermore, purifying selection may occur at different developmental stages and in different tissues (germline versus soma) from selfish selection, creating an oscillation in the relative levels of the two genomes when comparing different developmental stages, without changing the ratio of the genomes when comparing different generations (figure 4*b*) [111,127].

The nuclear background can influence the strength of various types of selection, and thus alter the outcome of mtDNA competition. This is because the nuclear genome encodes nearly all of the proteins in mitochondria, as well as external regulators of mitochondrial biogenesis, dynamics and mitophagy/turnover. Differences in the nuclear genome can re-define the functional OXPHOS capacity of mtDNA and determine whether or not a variant is a detrimental mutation: one mtDNA variant may result in deficient OXPHOS capacity in one nuclear background but not in another owing to differences in the nuclear-encoded complex proteins [128,129]. The strength of purifying selection can depend on the severity of mismatch. Selfish transmission of certain mtDNA variants may only manifest in a given nuclear background as particular isoforms of nuclear genes are required to allow them to replicate or transmit better. Tissues with different energy demands may have preferences for mitochondrial genomes with certain metabolic rates, energy expenditures or replication features [130–132]. Other changes, such as temperature and age, may also impact mtDNA competition, probably through altering nuclear transcriptional profiles [133–135].

The nuclear influence on mtDNA competition has been shown in a number of studies. In *C. briggsae*, the occurrence

of mtDNA with large deletions varies between different strains with different nuclear genomes [103]. In *D. subobscura*, the abundance of the 5 kb deletion-bearing molecule was stable across generations, but changed during backcrosses to a different nuclear genotype [136]. In *D. melanogaster*, the level of *D. yakuba* mtDNA was initially stabilized at 5% when paired with *mt:Col^{ts}*, but in another nuclear background it stabilized at 20% [111]. In mice, tissue-specific segregation of heteroplasmy has been reported in a number of studies [137–139]. Furthermore, certain human alleles are selected for at specific nucleotide positions in specific tissues as individuals age [12].

# 5. Heteroplasmy and mitochondrial replacement therapy

The last 5–10 years has been an exciting time for MRT, as fundamental scientific discoveries have significantly advanced the clinical strategies to prevent the transmission of pathogenic mitochondrial mutations. In 2016, the first 'three-parent boy' was born in Mexico [55], and in early 2017 the first 'three-parent girl' was born in Ukraine. In the UK, three-parent babies could be born this year, as two cases have been approved by the UK's Human Fertilization and Embryology Authority to take place at the Newcastle Fertility Centre.

However, MRT has raised a number of ethical and safety concerns. Although it has been argued that MRT poses no greater risk of mito-nuclear incompatibility in humans than normal reproduction [140,141], matching the parents' nuclear genome with the donor's mitochondrial genome could be considered to minimize potential effects of mito-nuclear interactions observed in cybrid studies and genetic rescues [142–144].

Another safety concern is the carryover of mutant mtDNA from the original mother's egg. Historically, embryos carrying mtDNA from both a donor and the mother were created by cytoplasmic transfer, which was developed to enable women with impaired fertility to bear children. Although mitochondrial transfer was not the primary objective at the time, 5–15% of ooplasm from unfertilized oocytes is transferred to the recipient oocyte during this process, thus creating babies with multiple mitochondrial genotypes. Analysis of the mtDNA from offspring produced using cytoplasmic transfer confirmed the presence of donor mtDNA. Between the late 1990s and early 2000s, cytoplasmic transfer resulted in over 30 live births in the USA [145]. Currently, MRT can be performed by either maternal spindle transfer before fertilization or by pro-nuclear transfer after fertilization, and both methods lead to some carryover [48–50,54]. For example, oocytes and embryos produced from maternal spindle transfer by Tachibana *et al.* [48] had a mean carryover of 0.5%, while embryos produced from pro-nuclear transfer by Craven *et al.* [49] had a mean carryover of 1.8%. In addition, the first boy born from MRT via spindle transfer contained 2–9% maternal mutant mtDNA in tissues examined (hair follicles, circumcised foreskin and umbilical blood) [55]. Recently, polar body transfer has been suggested as an alternative approach to reduce mtDNA carryover [54,146,147], as polar bodies contain few mtDNA copies. Nevertheless, an average carryover of 0.26% in blastocysts has been observed [54].

Even a small trace of carried-over mutant mtDNA could reach the disease-causing threshold level in specific tissues later in life. This can occur through genetic drift, as has been observed when passaging human pluripotent stem cell lines derived from blastocysts created by MRT [52]. Reversion to maternal haplotype can occur more rapidly when the maternal genome has a replicative advantage. For example, in a study where spindle transfers were carried out between healthy human oocytes with preselected mtDNA haplotypes, two out of 15 blastocyst-derived embryonic stem cell lines reverted to the maternal haplotype [53]. These two cell lines were created by transfer events that mixed a maternal haplotype U5a with a donor haplotype H1b (differ by 33 SNPs). Sibling cell lines generated that mixed the same maternal haplotype with a different donor haplotype V3 (also differ by 33 SNPs) did not show reversion, suggesting that reversion is specific to a certain combination of haplotypes in the maternal nuclear background. The starting abundance of the maternal mtDNA was less than 1%, but it reached 81% and 94%, respectively, after two or three passages. Further passaging resulted in a complete loss of donor mtDNA. This reversion also occurred during stem cell differentiation, raising the possibility that reversal to the mutant mtDNA may occur in some MRT children. The group identified a polymorphism within the control region of the maternal haplotype and suggested that this polymorphism could enhance replication priming and thus the proliferation of the maternal genome when paired with H1b [53]. This is very similar to the selfish selection described earlier in *Drosophila*, where genomes with a certain non-coding region showed a transmission advantage regardless of their OXPHOS function [111].

Although it is not known whether mtDNA shifts in embryonic stem cell lines will truly reflect those in the developing embryo, precautions ought to be taken to minimize the reversion after MRT. In humans, there are at least 25 major mitochondrial haplotypes, each containing many subclades that differ significantly for their control region [148]. If we can make sure that the donor's mitochondrial genotype not only is compatible with the nuclear genome, but also has a selective advantage in replication or transmission, it would have the advantage of fully outcompeting the maternal genome, even if the carryover level is high. For that, we need to know more about how the outcome of competition is determined in order to know which genome will have a competitive advantage. In particular, we need to identify sequences in mtDNA that can confer replicative advantage to certain mitochondrial genomes and understand how changes in the nuclear genome can influence the outcome of mtDNA competition [149]. This is relevant not only to MRT but also to cytoplasmic transfer, which, although abandoned in the USA because of uncertainties about its safety and effective benefit, is still commercially available in IVF clinics in numerous countries worldwide [150].

# 6. Conclusion and future perspectives

It is encouraging to see that powerful tools and animal models have been established to reveal how different forces influence the transmission of mtDNA, given that selective transmission influences mitochondrial disease and guides mtDNA evolution. Moreover, recent advances in genome

royalsocietypublishing.org/journal/rsob    *Open Biol.* **9**: 180267

royalsocietypublishing.org/journal/rsob Open Biol. 9: 180267

editing based on mitochondrially targeted transcription activator-like effector nucleases and zinc finger nucleases have allowed selective elimination of pathogenic mutations in mouse germ cells [151] and somatic tissues [152,153], and in induced pluripotent stem cells derived from patients with mitochondrial encephalomyopathy, lactic acidosis and stroke-like episodes (MELAS) [154]. These technological breakthroughs will certainly widen the therapeutic options in the near future. Nevertheless, the study of mtDNA is far from exhausted and the management of mitochondrial diseases has lagged behind the genetic revolution. In the future, we need to gain a better understanding of what and how sequence differences in mtDNA give a selfish transmission advantage and how the nuclear genome modulates purifying selection to safeguard the organismal investment in mitochondrial genes. To answer these questions, we need to pursue even more basic questions such as how mtDNA replication is controlled and how the genome segregates. Furthermore, essential aspects of mitochondrial biology that were once thought fundamental and universal, such as the lack of recombination [36–43] and strict maternal inheritance [17–23], must now be revisited with new tools and systems that provide higher detection power.

Data accessibility. This article has no additional data.
Competing interests. We declare we have no competing interests.
Funding. Funding for this work was provided by the Wellcome Trust (grant no. 202269/Z/16/Z).

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
