## [Reviewer comments · Open Biology]

Review History

RSOB-18-0267.R0 (Original submission)

Review form: Reviewer 1

Recommendation

Major revision is needed (please make suggestions in comments)

Are each of the following suitable for general readers?

- a) **Title**
Yes
- b) **Summary**
Yes
- c) **Introduction**
Yes

Is the length of the paper justified?

Yes

Should the paper be seen by a specialist statistical reviewer?

No

Is it clear how to make all supporting data available?

Not Applicable

Is the supplementary material necessary; and if so is it adequate and clear?

Not Applicable

Do you have any ethical concerns with this paper?

No

Comments to the Author

Review of "A battle for transmission: the cooperative and selfish mitochondrial genome" by Kulcnika and Ma.

In the current manuscript, the authors provide a review of mitochondrial transmission in light of selfish mitonuclear conflict: the idea that mt genomes with detrimental organismal effects may persist due to a replication advantage. They provide details on theory and case studies of how nuclear-encoded mechanisms act to eliminate mt variants with detrimental organismal effects, as well as complementary details on how specific mt variants may increase their own transmission at the expense of the organism. Finally, these evolutionary dynamics are considered in light of emerging MRT practices, especially with regard to how carryover effects of deleterious genomes coupled with selfish replication may decrease MRT efficacy.

Overall, I found the manuscript interesting and enlightening. The authors provide a succinct review of many key relevant points in this field and highlight many of the key previous studies (including those by Ma). However, one persistent shortcoming throughout is confusion over the term "selfish" and mixing of various terms associated with "selection". In particular, a mt genome is only "selfish" if it has a transmission advantage (at the genome or organelle level) while at the same time causing a decrease in organismal fitness. For example, in the paragraph at the top of page 7, it is stated that "Of note, selfish selection can be neutral to the host when the selfish drive is not linked to detrimental mutations." If an mt variant has a replication advantage but does not result in reduced organismal fitness, it is not selfish. Below, I have noted places throughout the manuscript where this is a problem, but I would encourage the authors to edit the entire manuscript with this in mind. I have also noted many other minor points that would improve the manuscript, including stating conflicts associated with Mother's curse more explicitly, considering mt diversity outside of animals (or stating that the review is animal focused), and noting the possibility of selfish transmission and replication advantage at the organelle level.

There are no line numbers to reference, so I have referred to sections, paragraphs, and sentences as specifically as possible.

Specific comments

- 1) Abstract first line - remove "evolutionary"
- 2) "Host" is used throughout the abstract and at many points in the manuscript, but I don't think

this is quite appropriate considering that the mt predecessor is long gone and the “host” is really a chimera of the original two partners. Consider switching this to “organism” or “individual”.

3) Abstract – last line. Considering that MRT plays a major role in the manuscript, I suggest defining it more explicitly and in more detail in the abstract for readers not familiar with it.

Background

4) 1st paragraph. It is implied in these first lines that the original mt host cell was an early eukaryote, although this is contentious and many think it was an archaeon (e.g., Martin’s hydrogen hypothesis). I suggest rephrasing to something a bit more ambiguous such as “acquired early in eukaryotic evolution” to avoid going into this controversy.

5) Similarly, the idea that the mt genome provided a “monumental” upgrade to the energy supply of early eukaryotes is contentious. If this statement remains, I suggest at least citing some of Nick Lane’s papers espousing this idea (e.g., Lane 2014 Cold Spring Harb Perspect Biol), or adding a caveat such as “possibly” and also citing some opposing viewpoints (e.g., Lynch and Marinov 2015 PNAS).

6) Figure 1 – while I agree that the human mt genome is largely representative of mammals, there is an incredible amount of diversity in mt genome content and organization in insects and in eukaryotes more broadly. I suggest the authors acknowledge this explicitly and make the concession that the review is focused mainly on bilaterian animal mt genomes. Possibly also cite Smith and Keeling 2015 for those interested in examining this diversity.

7) In Fig 1 caption – about how many nuclear grey genes are there in the ETC compared with the 13 mt genes?

8) “As such, mtDNA does not have to face any heredity competitors” – make it clear that heteroplasmy can arise via inheritance of a heteroplasmic pool of mtDNA from the mother as well as mutation and biparental inheritance. However, the point that maternal inheritance acts to limit heteroplasmy is well taken.

9) “Theoretically the high mutation rate” – again, this is biased to bilaterian animals. Corals and most plants actually have low mt mutation rates. Make this clear.

10) “paternal leakage” – probably needs to be defined and contrasted with “biparental inheritance”

11) The last sentence here on DUI molluscs is too brief to give readers a proper introduction. I suggest either removing mention of DUI or adding a few sentences, making it clear that this system is still being investigated, but it seems that heteroplasmy is common at different levels in at least some male somatic (but not gonadal) tissues.

12) “Heteroplasmy represents a dynamic and constantly changing...” these few sentences need to have citations backing them up and should also be a bit less definitive, adding caveats such as “can” and “may”, given that the rest of the paper is citing examples where mt genome transmission is anything but random.

13) Top of page 3 – change “selections” to “selection”

14) Next paragraph – change “recently emerged” to “emerging”

15) Last paragraphs in this section and in Fig. 2 caption – change “selections” to “levels of selection”

16) Fig 2 – I think the cartoon doesn’t represent “random segregation” as I understand it. I think this refers to the random transmission of organelles to the next generation, not genomes. I suggest changing it so that the same 3 organelles are present in the next generation, but at variable proportions. Alternatively, it might be useful to divide into “organelle” vs “genome” levels and show how proportions can change within a cell at these different levels. Finally, the “bottleneck” on the left corresponds to a new generation, but it is unclear whether this applies across the figure. I suggest adding a horizontal line to make it clear where F0 transitions to F1.

Host-beneficial purifying selection

17) See previous comments about rephrasing the title of this section – maybe to “Selection for organismal fitness”

18) “The nuclear genome is thus interested in passing on...” I think this sentence is a bit misleading and would be phrased better in terms of mtDNA that benefits organismal fitness

19) “Given that the nuclear genome...” change “all” in this sentence to “most” or rephrase to be clear only non-OXPHOS activities are being considered

20) 3rd paragraph “This reduction in mtDNA copy number...” in this sentence do the “large shifts” tend to be toward less or more heteroplasmy?

21) 4th paragraph “In *Drosophila*, there is also evidence linking...” in this sentence the main point is that the nucleus will tend to replicate functional mtDNA copies over deleterious ones. However, a common nuclear response to OXPHOS deficiency may be to increase mtDNA copy number. Therefore, defective mtDNA may actually be propagated preferentially by the nucleus. This concept needs to be mentioned. There is some empirical evidence for this in the Gitschlag et al. reference that is already cited, and some evidence based on modeling as well (Capps et al. 2003 *J. Theor. Biol.*; Chinnery et al. 1999 *Am. J. Hum. Genet.*; Tam et al. 2015 *Plos Comput. Biol.*)

22) In the next line, it is not clear that the mt:COI mutant is actually selfish. It is deleterious, but does it have a transmission advantage? This should be discussed here when it is first introduced.

23) 5th paragraph “In a study using *D. melanogaster* heteroplasmic for both...” does “deletion molecules” refer to organelles or genomes? If the latter, rephrase to “deletion-bearing mt genomes”.

24) 6th paragraph – similarly here, it is not clear whether these mt mutations are selfish or just deleterious. Also in this paragraph, there is a lot of mention of purifying selection not taking place. It’s not that purifying selection is absent (if the mutations are deleterious than by definition they are evolving under purifying selection), it just might not be strong/efficient enough to eliminate them.

Selfish selection

25) 2nd paragraph first/second sentence – this is another sentence in which the muddled definition of “selfish” comes through. By definition a mt genome must be deleterious to organismal fitness to be selfish.

26) 3rd paragraph "A recent study suggested that uaDf5..." Again, mention here that it appears this mutation "hijacks" the replication control mechanisms and proliferates because it reduces OXPHOS function.

27) 4th paragraph - I think a key takeaway here is whether a mt genome has a replication advantage or not depends on what other mt genomes its competing against. Therefore, while the mt:COI mutation may be deleterious in any context, it only has a replication advantage in certain situations. The recombination result is very interesting and also suggests that the replication advantage mutation and the deleterious mutation are not necessarily the same in all cases, but may be effectively linked due to the non-recombination of the mt genome. Make this clear - that the "selfish" quality of an mt genome may be due to multiple linked variants.

28) 5th paragraph - is a takeaway here that nuclear background also matters as to whether a mt genome has a replication advantage or is deleterious? "Interestingly, mtDNA from D. mauritiana..." in this sentence it is not clear that the foreign mtDNA is selfish, does it have any deleterious effects?

29) 6th paragraph - as mentioned above, the first sentence muddies the definition of selfish - non-deleterious mt genomes cannot be selfish.

The interplay of different selections at multiple levels

30) 1st paragraph - here and throughout, consider whether the terms "purifying" and "selfish" selection might be better replaced with "organismal selection" and "selection for replication" or something similar that indicates the difference in levels of selection.

31) 1st paragraph - paternal leakage is quite common in some plant lineages, and has even been observed at high levels in Drosophila (e.g., Birky 1995 PNAS; James and Ballard 2003 Genetics)

32) 2nd paragraph - "This can happen if the selfish drive only..." does "small" in this sentence imply that if the variant gives a very strong replication advantage they will never reach high levels? It is a bit confusing.

33) 2nd paragraph - "Selfish selection mainly occurs at the genome level, whereas..." it is unclear to me why selfish conflict could not occur at the organelle level as well as the genome level. Couldn't a mt variant cause mitochondrial organelle proliferation, but result in less fit organelles as well? Possibly by manipulating membrane or fission cycles? Is there evidence for this?

34) Fig 3. - at the top right, I suggest adding a comparative panel with a male beneficial mutation to show that because males do not transmit mt genomes, both mt genomes will reach the same level in the next generation regardless of their effects on males.

35) 3rd paragraph beginning with "The nuclear background can influence the strength of purifying..." The entire point of this paragraph was unclear to me. It is unclear what is meant by "levels of proteins" "nuclear isoforms" or "this would reduce the need and strength of purifying selection". There may be important points in this paragraph, but it was largely confusing to me what the authors were trying to get across.

Heteroplasmy and mitochondrial replacement therapy

36) 1st paragraph - I think there are actually a handful of three-parent babies that are now teenagers. These procedures were done illegally I believe and are not well-documented, but I

think the authors should look into this and make this sentence reflect the history or MRT, although I'm sure it's a bit murky. See this popular article:
<https://slate.com/technology/2016/02/three-parent-babies-have-been-here-since-the-late-90s.html>

37) Last paragraph – “If we can make sure that the...” the idea of selecting a donor with an mt genome that will have a replication advantage is really interesting, but you might want to reiterate here that mitonuclear interactions might want to also be considered in selecting a donor. For example, an ideal donor would have a mt genome that is compatible with the patient's nuclear DNA and has a replication advantage against its defective mtDNA.

38) One thing that isn't clear in this last section is why does carryover occur at all? Is this just because the laboratory techniques are imperfect and a bit of the patient's cytoplasm with some mitochondria is always taken by accident? Would there be a way to eradicate the patient's cytoplasm/mt genomes prior to adding the donor cytoplasm?

Review form: Reviewer 2

Recommendation

Accept with minor revision (please list in comments)

Are each of the following suitable for general readers?

- a) **Title**
No
- b) **Summary**
No
- c) **Introduction**
No

Is the length of the paper justified?

Yes

Should the paper be seen by a specialist statistical reviewer?

No

Is it clear how to make all supporting data available?

No

Is the supplementary material necessary; and if so is it adequate and clear?

No

Do you have any ethical concerns with this paper?

No

Comments to the Author

Title:

A battle for transmission: the cooperative and selfish mitochondrial genome

Authors:

Anna Klucnika and Hansong Ma

Summary:

In this manuscript, Klucnika and Ma review selfish behavior of mitochondrial genomes in bilaterian animals, with especial focus on human mitochondrial dynamics. The review discusses a variety of related topics, including heteroplasmy, selection, drift, and bottlenecks in the oocyte. Notably, the evidence described, while thoroughly researched, is restricted to bilaterian animals. This choice has the unintended effects of both limiting the topic to a relatively narrow section of mitochondrial diversity, and trying to pigeonhole mitochondrial biology of nematodes and insects onto humans.

Overall, I found the topic to be of broad interest to the readership of Open Biology, and if it is recast and re-titled to focus more narrowly on heteroplasmy in bilaterians, to be quite well done. I also think the manuscript could use more careful editing, as there were several language usages that I found odd and unclear. Personally, I find the focus on MRT to be less interesting than the dynamics of heteroplasmy, but the authors do provide a novel perspective on it. Assuming the authors can narrow the scope of their review and address a couple other minor issues, I think this manuscript would make for a nice contribution to Open Biology.

Major Concerns:

1. Framing of the manuscript. The manuscript in its current state is framed as a broad look into mitochondrial biology; however, little if any of the references discuss mitochondrial biology outside of mammals, nematodes, and insects. This choice to ignore the vast majority of mitochondrial biology while trying to draw broad conclusions is odd. Much of what the authors are trying to discuss has been researched extensively in other taxa, especially plants, but no credit is given to the work done in that field. For example, mitochondrial genomes often experience rampant recombination, varying degrees of biparental inheritance exist in non-bilaterian lineages, and mutation rate varies orders of magnitude across taxonomic groups, even within metazoa. This, as one might imagine, has profound consequences for mitochondrial biology, especially with respect to the presumed mutational meltdown mitochondrial genomes are supposed to experience. Accordingly, the authors need to either broaden their literature scope considerably or restrict their observations to bilaterian animals. I have also included a couple citations below whose inclusion would help improve the manuscript, even if the focus becomes narrower.

2. Emphasis on purifying selection/genetic drift. The section describing the forces that can affect mitochondrial genome evolution and selfish mitochondrial genome proliferation was overly simplistic. While purifying selection can and does play a major role in mitochondrial genome evolution, the authors neglect entirely the role of positive selection in mitochondrial biology. A poignant example can be glimpsed in mitochondria present in populations that have migrated to low-oxygen conditions (e.g., high elevation) relative to their ancestors and/or close relatives. While the mitochondrial genome is small and rates of recombination are low, beneficial mutations can still experience positive selection, especially when mitochondria are poorly adapted to local environmental conditions (i.e., when fitness effects of mutations are large).

While much of the manuscript was well done, I have a couple additional minor comments that the authors might consider to improve their manuscript.

- Page 2, when describing the genetic content of mitochondrial genomes. Note that this section applies exclusively to bilaterians. Mitochondrial genomes of non-bilaterians can differ quite dramatically from those of the systems that are the focus of the present study (e.g., SDH is commonly still mitochondrially encoded in many plant lineages)
- Similarly, the point about high mutation rates does not apply beyond bilaterians. See Lavrov and Pett for a nice review on metazoan mitochondrial genome diversity.

- First paragraph of page 3 – “so the genome is under the strong influence of genetic drift”. I think a lot of mitochondrial researchers would object strongly to this, absent some contextual justification. See Konrad et al. 2017 for a contradictory opinion.
- There is a repeated use of the term “selections”. I believe it would be better to say “types of selection”.
- Page 4 “The nuclear genome is thus interested in...”. This is another problematic sentence formulation. The nuclear genome is not interested in anything. Nuclear genomes that successfully coordinate mitochondrial expression/replication/transmission have higher fitness than nuclear genomes that do not.
- Cite Holland et al. 2018 on page 4 somewhere.
- “Limited recombination has little power to remove de novo mutations”. I do not believe this statement to be factually correct. See Neiman and Taylor 2009. Only a little bit of recombination is necessary to reset Muller’s Ratchet.
- I think the phrase “selfish selection” is odd. I would change to “Selfish proliferation”. While selection can act at the level of the replicator (i.e., rapid/efficiently replicating mtDNA molecules have a fitness advantage over slowly replicating mtDNA molecules), I think the phrasing here sparks of some active role of the molecule in its own selection. The term “proliferation” better encapsulates the phenomenon.
- The lack of any information about cytoplasmic male sterility in plants seems a glaring omission in the selfish element section. This is a major reason why I think the manuscript needs to be re-framed/narrowed in scope.
- Remove the reference to unpublished data from the top part of page 9. Given that it is unpublished, the audience cannot properly evaluate the quality of the evidence supporting the point. The resulting conclusion should therefore be removed until the data can be properly viewed.
- The description of mitonuclear incompatibility and subsequent compensatory coevolution is not correctly described. See Sloan et al. 2017 for a better description of the dynamics at play.
- In the MRT section, I think it would be worth citing Havird et al 2016.

Recommended literature:

Barr et al., 2005. Inheritance and recombination of mitochondrial genomes in plants, fungi and animals. *New Phytol.* 168:39-50.

Havird et al., 2016. Sex, mitochondria, and genetic rescue. *TREE* 31: 96-99.

Holland et al., 2018. Deep-Coverage MPS Analysis of Heteroplasmic Variants within the mtGenome Allows for Frequent Differentiation of Maternal Relatives. *Genes.* 9: 124.

Konrad et al., 2017. Mitochondrial Mutation Rate, Spectrum and Heteroplasmy in *Caenorhabditis elegans* Spontaneous Mutation Accumulation Lines of Differing Population Size. *MBE.* 34: 1319-1334.

Lavrov and Pett 2016. *GBE.* Animal mitochondrial DNA as we do not know it: mt-genome organization and evolution in nonbilaterian lineages. 8: 2896-2913.

Neiman and Taylor 2009. The causes of mutation accumulation in mitochondrial genomes. *Proc Roy Soc B.* 276: 1201-1209.

Sloan et al., 2017. The on-again off-again relationship between mitochondria and species boundaries.

Decision letter (RSOB-18-0267.R0)

04-Feb-2019

Dear Dr Ma

We are pleased to inform you that your manuscript RSOB-18-0267 entitled "A battle for transmission: the cooperative and selfish mitochondrial genome" has been accepted by the Editor for publication in Open Biology. The reviewer(s) have recommended publication, but also suggest some minor revisions to your manuscript. Therefore, we invite you to respond to the reviewer(s)' comments and revise your manuscript.

Please submit the revised version of your manuscript within 14 days. If you do not think you will be able to meet this date please let us know immediately and we can extend this deadline for you.

- 1) A text file of the manuscript (doc, txt, rtf or tex), including the references, tables (including captions) and figure captions. Please remove any tracked changes from the text before submission. PDF files are not an accepted format for the "Main Document".
- 2) A separate electronic file of each figure (tiff, EPS or print-quality PDF preferred). The format should be produced directly from original creation package, or original software format. Please note that PowerPoint files are not accepted.
- 3) Electronic supplementary material: this should be contained in a separate file from the main text and meet our ESM criteria (see <http://royalsocietypublishing.org/instructions-authors#question5>). All supplementary materials accompanying an accepted article will be treated as in their final form. They will be published alongside the paper on the journal website and posted on the online figshare repository. Files on figshare will be made available approximately one week before the accompanying article so that the supplementary material can be attributed a unique DOI.

Online supplementary material will also carry the title and description provided during submission, so please ensure these are accurate and informative. Note that the Royal Society will not edit or typeset supplementary material and it will be hosted as provided. Please ensure that

the supplementary material includes the paper details (authors, title, journal name, article DOI). Your article DOI will be 10.1098/rsob.2016[*last 4 digits of e.g. 10.1098/rsob.20160049*].

4) A media summary: a short non-technical summary (up to 100 words) of the key findings/importance of your manuscript. Please try to write in simple English, avoid jargon, explain the importance of the topic, outline the main implications and describe why this topic is newsworthy.

Images

Data-Sharing

It is a condition of publication that data supporting your paper are made available. Data should be made available either in the electronic supplementary material or through an appropriate repository. Details of how to access data should be included in your paper. Please see <http://royalsocietypublishing.org/site/authors/policy.xhtml#question6> for more details.

Data accessibility section

Sincerely,

The Open Biology Team

<mailto:openbiology@royalsociety.org>

Editors comment:

Hansong - please attend to all of the referees' comments - many thanks

David

Reviewer(s)' Comments to Author:

Referee: 1

Comments to the Author(s)

Review of "A battle for transmission: the cooperative and selfish mitochondrial genome" by Kulcnika and Ma.

In the current manuscript, the authors provide a review of mitochondrial transmission in light of selfish mitonuclear conflict: the idea that mt genomes with detrimental organismal effects may persist due to a replication advantage. They provide details on theory and case studies of how nuclear-encoded mechanisms act to eliminate mt variants with detrimental organismal effects, as well as complementary details on how specific mt variants may increase their own transmission at the expense of the organism. Finally, these evolutionary dynamics are considered in light of

emerging MRT practices, especially with regard to how carryover effects of deleterious genomes coupled with selfish replication may decrease MRT efficacy.

Overall, I found the manuscript interesting and enlightening. The authors provide a succinct review of many key relevant points in this field and highlight many of the key previous studies (including those by Ma). However, one persistent shortcoming throughout is confusion over the term “selfish” and mixing of various terms associated with “selection”. In particular, a mt genome is only “selfish” if it has a transmission advantage (at the genome or organelle level) while at the same time causing a decrease in organismal fitness. For example, in the paragraph at the top of page 7, it is stated that “Of note, selfish selection can be neutral to the host when the selfish drive is not linked to detrimental mutations.” If an mt variant has a replication advantage but does not result in reduced organismal fitness, it is not selfish. Below, I have noted places throughout the manuscript where this is a problem, but I would encourage the authors to edit the entire manuscript with this in mind. I have also noted many other minor points that would improve the manuscript, including stating conflicts associated with Mother’s curse more explicitly, considering mt diversity outside of animals (or stating that the review is animal focused), and noting the possibility of selfish transmission and replication advantage at the organelle level.

There are no line numbers to reference, so I have referred to sections, paragraphs, and sentences as specifically as possible.

Specific comments

- 1) Abstract first line – remove “evolutionary”
- 2) “Host” is used throughout the abstract and at many points in the manuscript, but I don’t think this is quite appropriate considering that the mt predecessor is long gone and the “host” is really a chimera of the original two partners. Consider switching this to “organism” or “individual”.
- 3) Abstract – last line. Considering that MRT plays a major role in the manuscript, I suggest defining it more explicitly and in more detail in the abstract for readers not familiar with it.

Background

- 4) 1st paragraph. It is implied in these first lines that the original mt host cell was an early eukaryote, although this is contentious and many think it was an archaeon (e.g., Martin’s hydrogen hypothesis). I suggest rephrasing to something a bit more ambiguous such as “acquired early in eukaryotic evolution” to avoid going into this controversy.
- 5) Similarly, the idea that the mt genome provided a “monumental” upgrade to the energy supply of early eukaryotes is contentious. If this statement remains, I suggest at least citing some of Nick Lane’s papers espousing this idea (e.g., Lane 2014 Cold Spring Harb Perspect Biol), or adding a caveat such as “possibly” and also citing some opposing viewpoints (e.g., Lynch and Marinov 2015 PNAS).
- 6) Figure 1 – while I agree that the human mt genome is largely representative of mammals, there is an incredible amount of diversity in mt genome content and organization in insects and in eukaryotes more broadly. I suggest the authors acknowledge this explicitly and make the concession that the review is focused mainly on bilaterian animal mt genomes. Possibly also cite Smith and Keeling 2015 for those interested in examining this diversity.

7) In Fig 1 caption – about how many nuclear grey genes are there in the ETC compared with the 13 mt genes?

8) “As such, mtDNA does not have to face any heredity competitors” – make it clear that heteroplasmy can arise via inheritance of a heteroplasmic pool of mtDNA from the mother as well as mutation and biparental inheritance. However, the point that maternal inheritance acts to limit heteroplasmy is well taken.

9) “Theoretically the high mutation rate” – again, this is biased to bilaterian animals. Corals and most plants actually have low mt mutation rates. Make this clear.

10) “paternal leakage” – probably needs to be defined and contrasted with “biparental inheritance”

11) The last sentence here on DUI molluscs is too brief to give readers a proper introduction. I suggest either removing mention of DUI or adding a few sentences, making it clear that this system is still being investigated, but it seems that heteroplasmy is common at different levels in at least some male somatic (but not gonadal) tissues.

12) “Heteroplasmy represents a dynamic and constantly changing...” these few sentences need to have citations backing them up and should also be a bit less definitive, adding caveats such as “can” and “may”, given that the rest of the paper is citing examples where mt genome transmission is anything but random.

13) Top of page 3 – change “selections” to “selection”

14) Next paragraph – change “recently emerged” to “emerging”

15) Last paragraphs in this section and in Fig. 2 caption – change “selections” to “levels of selection”

16) Fig 2 – I think the cartoon doesn’t represent “random segregation” as I understand it. I think this refers to the random transmission of organelles to the next generation, not genomes. I suggest changing it so that the same 3 organelles are present in the next generation, but at variable proportions. Alternatively, it might be useful to divide into “organelle” vs “genome” levels and show how proportions can change within a cell at these different levels. Finally, the “bottleneck” on the left corresponds to a new generation, but it is unclear whether this applies across the figure. I suggest adding a horizontal line to make it clear where F0 transitions to F1.

Host-beneficial purifying selection

17) See previous comments about rephrasing the title of this section – maybe to “Selection for organismal fitness”

18) “The nuclear genome is thus interested in passing on...” I think this sentence is a bit misleading and would be phrased better in terms of mtDNA that benefits organismal fitness

19) “Given that the nuclear genome...” change “all” in this sentence to “most” or rephrase to be clear only non-OXPHOS activities are being considered

20) 3rd paragraph “This reduction in mtDNA copy number...” in this sentence do the “large shifts” tend to be toward less or more heteroplasmy?

21) 4th paragraph “In *Drosophila*, there is also evidence linking...” in this sentence the main point is that the nucleus will tend to replicate functional mtDNA copies over deleterious ones. However, a common nuclear response to OXPHOS deficiency may be to increase mtDNA copy number. Therefore, defective mtDNA may actually be propagated preferentially by the nucleus. This concept needs to be mentioned. There is some empirical evidence for this in the Gitschlag et al. reference that is already cited, and some evidence based on modeling as well (Capps et al. 2003 *J. Theor. Biol.*; Chinnery et al. 1999 *Am. J. Hum. Genet.*; Tam et al. 2015 *Plos Comput. Biol.*)

22) In the next line, it is not clear that the mt:COI mutant is actually selfish. It is deleterious, but does it have a transmission advantage? This should be discussed here when it is first introduced.

23) 5th paragraph “In a study using *D. melanogaster* heteroplasmic for both...” does “deletion molecules” refer to organelles or genomes? If the latter, rephrase to “deletion-bearing mt genomes”.

24) 6th paragraph – similarly here, it is not clear whether these mt mutations are selfish or just deleterious. Also in this paragraph, there is a lot of mention of purifying selection not taking place. Its not that purifying selection is absent (if the mutations are deleterious than by definition they are evolving under purifying selection), it just might not be strong/efficient enough to eliminate them.

Selfish selection

25) 2nd paragraph first/second sentence – this is another sentence in which the muddled definition of “selfish” comes through. By definition a mt genome must be deleterious to organismal fitness to be selfish.

26) 3rd paragraph “A recent study suggested that *uaDf5*...” Again, mention here that it appears this mutation “hijacks” the replication control mechanisms and proliferates because it reduces OXPHOS function.

27) 4th paragraph – I think a key takeaway here is whether a mt genome has a replication advantage or not depends on what other mt genomes its competing against. Therefore, while the mt:COI mutation may be deleterious in any context, it only has a replication advantage in certain situations. The recombination result is very interesting and also suggests that the replication advantage mutation and the deleterious mutation are not necessarily the same in all cases, but may be effectively linked due to the non-recombination of the mt genome. Make this clear – that the “selfish” quality of an mt genome may be due to multiple linked variants.

28) 5th paragraph – is a takeaway here that nuclear background also matters as to whether a mt genome has a replication advantage or is deleterious? “Interestingly, mtDNA from *D. mauritiana*...” in this sentence it is not clear that the foreign mtDNA is selfish, does it have any deleterious effects?

29) 6th paragraph – as mentioned above, the first sentence muddies the definition of selfish – non-deleterious mt genomes cannot be selfish.

The interplay of different selections at multiple levels

30) 1st paragraph – here and throughout, consider whether the terms “purifying” and “selfish” selection might be better replaced with “organismal selection” and “selection for replication” or something similar that indicates the difference in levels of selection.

31) 1st paragraph – paternal leakage is quite common in some plant lineages, and has even been observed at high levels in *Drosophila* (e.g., Birky 1995 PNAS; James and Ballard 2003 Genetics)

32) 2nd paragraph – “This can happen if the selfish drive only...” does “small” in this sentence imply that if the variant gives a very strong replication advantage they will never reach high levels? It is a bit confusing.

33) 2nd paragraph – “Selfish selection mainly occurs at the genome level, whereas...” it is unclear to me why selfish conflict could not occur at the organelle level as well as the genome level. Couldn’t a mt variant cause mitochondrial organelle proliferation, but result in less fit organelles as well? Possibly by manipulating membrane or fission cycles? Is there evidence for this?

34) Fig 3. – at the top right, I suggest adding a comparative panel with a male beneficial mutation to show that because males do not transmit mt genomes, both mt genomes will reach the same level in the next generation regardless of their effects on males.

35) 3rd paragraph beginning with “The nuclear background can influence the strength of purifying...” The entire point of this paragraph was unclear to me. It is unclear what is meant by “levels of proteins” “nuclear isoforms” or “this would reduce the need and strength of purifying selection”. There may be important points in this paragraph, but it was largely confusing to me what the authors were trying to get across.

Heteroplasmy and mitochondrial replacement therapy

36) 1st paragraph – I think there are actually a handful of three-parent babies that are now teenagers. These procedures were done illegally I believe and are not well-documented, but I think the authors should look into this and make this sentence reflect the history or MRT, although I’m sure it’s a bit murky. See this popular article:
<https://slate.com/technology/2016/02/three-parent-babies-have-been-here-since-the-late-90s.html>

37) Last paragraph – “If we can make sure that the...” the idea of selecting a donor with an mt genome that will have a replication advantage is really interesting, but you might want to reiterate here that mitonuclear interactions might want to also be considered in selecting a donor. For example, an ideal donor would have a mt genome that is compatible with the patient’s nuclear DNA and has a replication advantage against its defective mtDNA.

38) One thing that isn’t clear in this last section is why does carryover occur at all? Is this just because the laboratory techniques are imperfect and a bit of the patient’s cytoplasm with some mitochondria is always taken by accident? Would there be a way to eradicate the patient’s cytoplasm/mt genomes prior to adding the donor cytoplasm?

Referee: 2

Comments to the Author(s)

Title:

A battle for transmission: the cooperative and selfish mitochondrial genome

Authors:

Anna Klucnika and Hansong Ma

Summary:

In this manuscript, Klucnika and Ma review selfish behavior of mitochondrial genomes in bilaterian animals, with especial focus on human mitochondrial dynamics. The review discusses a variety of related topics, including heteroplasmy, selection, drift, and bottlenecks in the oocyte. Notably, the evidence described, while thoroughly researched, is restricted to bilaterian animals. This choice has the unintended effects of both limiting the topic to a relatively narrow section of mitochondrial diversity, and trying to pigeonhole mitochondrial biology of nematodes and insects onto humans.

Overall, I found the topic to be of broad interest to the readership of Open Biology, and if it is recast and re-titled to focus more narrowly on heteroplasmy in bilaterians, to be quite well done. I also think the manuscript could use more careful editing, as there were several language usages that I found odd and unclear. Personally, I find the focus on MRT to be less interesting than the dynamics of heteroplasmy, but the authors do provide a novel perspective on it. Assuming the authors can narrow the scope of their review and address a couple other minor issues, I think this manuscript would make for a nice contribution to Open Biology.

Major Concerns:

1. Framing of the manuscript. The manuscript in its current state is framed as a broad look into mitochondrial biology; however, little if any of the references discuss mitochondrial biology outside of mammals, nematodes, and insects. This choice to ignore the vast majority of mitochondrial biology while trying to draw broad conclusions is odd. Much of what the authors are trying to discuss has been researched extensively in other taxa, especially plants, but no credit is given to the work done in that field. For example, mitochondrial genomes often experience rampant recombination, varying degrees of biparental inheritance exist in non-bilaterian lineages, and mutation rate varies orders of magnitude across taxonomic groups, even within metazoa. This, as one might imagine, has profound consequences for mitochondrial biology, especially with respect to the presumed mutational meltdown mitochondrial genomes are supposed to experience. Accordingly, the authors need to either broaden their literature scope considerably or restrict their observations to bilaterian animals. I have also included a couple citations below whose inclusion would help improve the manuscript, even if the focus becomes narrower.

2. Emphasis on purifying selection/genetic drift. The section describing the forces that can affect mitochondrial genome evolution and selfish mitochondrial genome proliferation was overly simplistic. While purifying selection can and does play a major role in mitochondrial genome evolution, the authors neglect entirely the role of positive selection in mitochondrial biology. A poignant example can be glimpsed in mitochondria present in populations that have migrated to low-oxygen conditions (e.g., high elevation) relative to their ancestors and/or close relatives. While the mitochondrial genome is small and rates of recombination are low, beneficial mutations can still experience positive selection, especially when mitochondria are poorly adapted to local environmental conditions (i.e., when fitness effects of mutations are large).

While much of the manuscript was well done, I have a couple additional minor comments that the authors might consider to improve their manuscript.

- Page 2, when describing the genetic content of mitochondrial genomes. Note that this section applies exclusively to bilaterians. Mitochondrial genomes of non-bilaterians can differ quite dramatically from those of the systems that are the focus of the present study (e.g., SDH is commonly still mitochondrially encoded in many plant lineages)
- Similarly, the point about high mutation rates does not apply beyond bilaterians. See Lavrov and Pett for a nice review on metazoan mitochondrial genome diversity.
- First paragraph of page 3 – “so the genome is under the strong influence of genetic drift”. I think a lot of mitochondrial researchers would object strongly to this, absent some contextual justification. See Konrad et al. 2017 for a contradictory opinion.

- There is a repeated use of the term “selections”. I believe it would be better to say “types of selection”.
- Page 4 “The nuclear genome is thus interested in...”. This is another problematic sentence formulation. The nuclear genome is not interested in anything. Nuclear genomes that successfully coordinate mitochondrial expression/replication/transmission have higher fitness than nuclear genomes that do not.
- Cite Holland et al. 2018 on page 4 somewhere.
- “Limited recombination has little power to remove de novo mutations”. I do not believe this statement to be factually correct. See Neiman and Taylor 2009. Only a little bit of recombination is necessary to reset Muller’s Ratchet.
- I think the phrase “selfish selection” is odd. I would change to “Selfish proliferation”. While selection can act at the level of the replicator (i.e., rapid/efficiently replicating mtDNA molecules have a fitness advantage over slowly replicating mtDNA molecules), I think the phrasing here sparks of some active role of the molecule in its own selection. The term “proliferation” better encapsulates the phenomenon.
- The lack of any information about cytoplasmic male sterility in plants seems a glaring omission in the selfish element section. This is a major reason why I think the manuscript needs to be re-framed/narrowed in scope.
- Remove the reference to unpublished data from the top part of page 9. Given that it is unpublished, the audience cannot properly evaluate the quality of the evidence supporting the point. The resulting conclusion should therefore be removed until the data can be properly viewed.
- The description of mitonuclear incompatibility and subsequent compensatory coevolution is not correctly described. See Sloan et al. 2017 for a better description of the dynamics at play.
- In the MRT section, I think it would be worth citing Havird et al 2016.

Recommended literature:

Barr et al., 2005. Inheritance and recombination of mitochondrial genomes in plants, fungi and animals. *New Phytol.* 168:39-50.

Havird et al., 2016. Sex, mitochondria, and genetic rescue. *TREE* 31: 96-99.

Holland et al., 2018. Deep-Coverage MPS Analysis of Heteroplasmic Variants within the mtGenome Allows for Frequent Differentiation of Maternal Relatives. *Genes.* 9: 124.

Konrad et al., 2017. Mitochondrial Mutation Rate, Spectrum and Heteroplasmy in *Caenorhabditis elegans* Spontaneous Mutation Accumulation Lines of Differing Population Size. *MBE.* 34: 1319-1334.

Lavrov and Pett 2016. *GBE.* Animal mitochondrial DNA as we do not know it: mt-genome organization and evolution in nonbilaterian lineages. 8: 2896-2913.

Neiman and Taylor 2009. The causes of mutation accumulation in mitochondrial genomes. *Proc Roy Soc B.* 276: 1201-1209.

Sloan et al., 2017. The on-again off-again relationship between mitochondria and species boundaries.

Author's Response to Decision Letter for (RSOB-18-0267.R0)

See Appendix A.

Decision letter (RSOB-18-0267.R1)

19-Feb-2019

Dear Dr Ma

We are pleased to inform you that your manuscript entitled "A battle for transmission: the cooperative and selfish animal mitochondrial genomes" has been accepted by the Editor for publication in Open Biology.

Sincerely,

The Open Biology Team
mailto: openbiology@royalsociety.org

Appendix A

Reviewer(s)' Comments to Author:

Referee: 1

Comments to the Author(s)

Review of "A battle for transmission: the cooperative and selfish mitochondrial genome" by Kulcnika and Ma.

In the current manuscript, the authors provide a review of mitochondrial transmission in light of selfish mitonuclear conflict: the idea that mt genomes with detrimental organismal effects may persist due to a replication advantage. They provide details on theory and case studies of how nuclear-encoded mechanisms act to eliminate mt variants with detrimental organismal effects, as well as complementary details on how specific mt variants may increase their own transmission at the expense of the organism. Finally, these evolutionary dynamics are considered in light of emerging MRT practices, especially with regard to how carryover effects of deleterious genomes coupled with selfish replication may decrease MRT efficacy.

Overall, I found the manuscript interesting and enlightening. The authors provide a succinct review of many key relevant points in this field and highlight many of the key previous studies (including those by Ma). However, one persistent shortcoming throughout is confusion over the term "selfish" and mixing of various terms associated with "selection". In particular, a mt genome is only "selfish" if it has a transmission advantage (at the genome or organelle level) while at the same time causing a decrease in organismal fitness. For example, in the paragraph at the top of page 7, it is stated that "Of note, selfish selection can be neutral to the host when the selfish drive is not linked to detrimental mutations." If an mt variant has a replication advantage but does not result in reduced organismal fitness, it is not selfish. Below, I have noted places throughout the manuscript where this is a problem, but I would encourage the authors to edit the entire manuscript with this in mind. I have also noted many other minor points that would improve the manuscript, including stating conflicts associated with Mother's curse more explicitly, considering mt diversity outside of animals (or stating that the review is animal focused), and noting the possibility of selfish transmission and replication advantage at the organelle level.

We are delighted to learn that the reviewer found our manuscript interesting and enlightening. We also very much appreciate all of the comments from this reviewer. We agree with many of the suggested changes (see the detailed responses below), which have made the manuscript a much better piece of work. In particular, we have restricted our discussion to only animal mtDNA, added citations and rephrased some expressions to make our descriptions more accurate and thorough. However, we disagree with reviewer's statement that a mt genome is only "selfish" if it has a transmission advantage while at the same time causing a decrease in organismal fitness. According to Dictionary.com, which is the world's leading online resource for English definitions, synonyms, word origins and etymologies, "*selfish gene is a gene considered primarily as an element that tends to replicate itself in a population, whether or not it has a direct effect on the organism that carries it*". In addition, according to a number of reviews on selfish genetic elements (SGEs) (e.g. Werren 2011, PNAS), "*SGEs are defined as elements that enhance their own transmission relative to the rest of the individual's genome but are neutral or harmful to the*

individual as a whole. Therefore, we consider selfish mtDNA to be genomes inherited in a biased manner relative to the rest of the mtDNA pool without caring the organismal fitness. They don't have to bring in a negative impact on the organism to cause a genetic conflict, although that makes their selfishness more apparent. We believe that we have made this clear in the revised manuscript by stating the following: 1. With few constraints on replication and segregation of mtDNA, free-wheeling intra-organismal competition is likely to select for the best replicator, regardless of its OXPHOS output; 2. The occurrence of selfish selection is hard to detect in natural populations, as its consequence only becomes obvious when the selfish genome also possesses a detrimental mutation; 3. Of note, selfish selection can be neutral to the host when the selfish drive is not linked to detrimental mutations. We hope that we have convinced the reviewer on this issue.

There are no line numbers to reference, so I have referred to sections, paragraphs, and sentences as specifically as possible.

Specific comments

1) Abstract first line – remove “evolutionary”

The reviewer did not give the reason for removing this word. We do not quite understand why this word is not appropriate here and so we have not removed it.

2) “Host” is used throughout the abstract and at many points in the manuscript, but I don't think this is quite appropriate considering that the mt predecessor is long gone and the “host” is really a chimera of the original two partners. Consider switching this to “organism” or “individual”.

In the classic view, “host” means an animal or plant on or in which a parasite or commensal organism lives. However, from a gene-centric view, host has been used extensively to describe organisms that carrying various types of genetic elements. Indeed, many literatures have taken up this new meaning of “host”, including the *Genes in Conflict: The biology of selfish genetic elements*, a well-received book written by Burt and Triver 2006. Hence, we think that it is appropriate to use the word “host” here, as we consider the organism to be the host for both mitochondrial and nuclear genomes, not just for mtDNA. Nevertheless, we have replaced “host” with “organism/individual” in some of our descriptions to minimize misunderstanding.

3) Abstract – last line. Considering that MRT plays a major role in the manuscript, I suggest defining it more explicitly and in more detail in the abstract for readers not familiar with it.

We are grateful for reviewer's suggestion. However, after several attempts to define MRT in the abstract, we have decided not to include such information. This is because it is hard to explain MRT and the issue of carryover in a few sentences. After all, our main focus is on how different types of selection shape mtDNA competition.

Background

4) 1st paragraph. It is implied in these first lines that the original mt host cell was an

early eukaryote, although this is contentious and many think it was an archaeon (e.g., Martin's hydrogen hypothesis). I suggest rephrasing to something a bit more ambiguous such as "acquired early in eukaryotic evolution" to avoid going into this controversy.

We agree with the reviewer and have made the changes accordingly.

5) Similarly, the idea that the mt genome provided a "monumental" upgrade to the energy supply of early eukaryotes is contentious. If this statement remains, I suggest at least citing some of Nick Lane's papers espousing this idea (e.g., Lane 2014 Cold Spring Harb Perspect Biol), or adding a caveat such as "possibly" and also citing some opposing viewpoints (e.g., Lynch and Marinov 2015 PNAS).

We have kept the statement, and cited Nick Lane's work. Thank you for the suggestion.

6) Figure 1 – while I agree that the human mt genome is largely representative of mammals, there is an incredible amount of diversity in mt genome content and organization in insects and in eukaryotes more broadly. I suggest the authors acknowledge this explicitly and make the concession that the review is focused mainly on bilaterian animal mt genomes. Possibly also cite Smith and Keeling 2015 for those interested in examining this diversity.

We strongly agree with the reviewer and have acknowledged the diversity of mtDNA in the first paragraph. We also cited Smith and Keeling's 2015 paper published in PNAS.

7) In Fig 1 caption – about how many nuclear grey genes are there in the ETC compared with the 13 mt genes?

We have decided to leave out such information in Figure 1, as the exact number for different ETC varies from species to species (e.g. human vs *Drosophila*, and some of the subunits are putative (Garia et al 2017, Cell Reports)). Instead, we have cited a paper that has provided such information in the figure legend.

8) "As such, mtDNA does not have to face any heredity competitors" – make it clear that heteroplasmy can arise via inheritance of a heteroplasmic pool of mtDNA from the mother as well as mutation and biparental inheritance. However, the point that maternal inheritance acts to limit heteroplasmy is well taken.

We have changed the sentence to: "as such, maternal genomes do not have to face any heredity competitors from the male parent".

9) "Theoretically the high mutation rate" – again, this is biased to bilaterian animals. Corals and most plants actually have low mt mutation rates. Make this clear.

We have changed the title by adding "animal" in front of the "mitochondrial genome", so the review is focused only on animal mtDNA. We also replaced high mutation rate with constantly-occurring mutations to be more inclusive.

10) "paternal leakage" – probably needs to be defined and contrasted with "biparental inheritance"

We have changed the way that we describe paternal leakage.

11) The last sentence here on DUI molluscs is too brief to give readers a proper introduction. I suggest either removing mention of DUI or adding a few sentences, making it clear that this system is still being investigated, but it seems that heteroplasmy is common at different levels in at least some male somatic (but not gonadal) tissues.

We have added more descriptions to elaborate on DUI.

12) “Heteroplasmy represents a dynamic and constantly changing...” these few sentences need to have citations backing them up and should also be a bit less definitive, adding caveats such as “can” and “may”, given that the rest of the paper is citing examples where mt genome transmission is anything but random.

We have added relevant references and “can” in the first sentence.

13) Top of page 3 – change “selections” to “selection”

We have made the correction.

14) Next paragraph – change “recently emerged” to “emerging”

We have made the correction.

15) Last paragraphs in this section and in Fig. 2 caption – change “selections” to “levels of selection”

We have made the correction.

16) Fig 2 – I think the cartoon doesn't represent “random segregation” as I understand it. I think this refers to the random transmission of organelles to the next generation, not genomes. I suggest changing it so that the same 3 organelles are present in the next generation, but at variable proportions. Alternatively, it might be useful to divide into “organelle” vs “genome” levels and show how proportions can change within a cell at these different levels. Finally, the “bottleneck” on the left corresponds to a new generation, but it is unclear whether this applies across the figure. I suggest adding a horizontal line to make it clear where F0 transitions to F1.

In Figure 2, we represent heteroplasmy dynamics during both somatic and germline transmission. Firstly, we have drawn the mitochondria in different conformations to emphasise the dynamic network that these organelles form. We intend for the main focus to be on the mitochondrial genomes and have endeavored for these genomes to stand out in the cartoon. Secondly, we choose to make the generation transitions ambiguous because these dynamics also apply to somatic cell divisions. However, we appreciate the reviewer's comments as these highlight that these aims have not been made clear. We have therefore edited the figure legend to improve on this.

Host-beneficial purifying selection

17) See previous comments about rephrasing the title of this section – maybe to “Selection for organismal fitness”

We have rephrased the title to Selection for organismal fitness. Thank you for the suggestion.

18) “The nuclear genome is thus interested in passing on...” I think this sentence is a bit misleading and would be phrased better in terms of mtDNA that benefits organismal fitness

We have deleted the sentence to remove the misunderstanding.

19) “Given that the nuclear genome...” change “all” in this sentence to “most” or rephrase to be clear only non-OXPHOS activities are being considered

We have changed the phrasing to “nearly all” to correct the statement.

20) 3rd paragraph “This reduction in mtDNA copy number...” in this sentence do the “large shifts” tend to be toward less or more heteroplasmy?

It can be both ways. Reducing the segregation number can create sister cells with very different mtDNA mutation loads. If there is no selection at the cellular level, then a female can produce oocytes with a very high or very low levels of mitochondrial mutations.

21) 4th paragraph “In *Drosophila*, there is also evidence linking...” in this sentence the main point is that the nucleus will tend to replicate functional mtDNA copies over deleterious ones. However, a common nuclear response to OXPHOS deficiency may be to increase mtDNA copy number. Therefore, defective mtDNA may actually be propagated preferentially by the nucleus. This concept needs to be mentioned. There is some empirical evidence for this in the Gitschlag et al. reference that is already cited, and some evidence based on modeling as well (Capps et al. 2003 J. Theor. Biol.; Chinnery et al. 1999 Am. J. Hum. Genet.; Tam et al. 2015 Plos Comput. Biol.)

For the *Drosophila* case, preferential replication of functional mtDNA does not lead to an increase in total mtDNA copy number. Furthermore, despite being unhealthy, the total mtDNA copy number of flies that are homoplasmic for *mt:Col^{ts}* is the same as flies with wild-type mtDNA. However, we agree that this is an important concept to mention. Hence, we have added it in the “selfish selection” chapter, after discussing the *uaDF5* genome as suggested by the this reviewer later.

22) In the next line, it is not clear that the *mt:COI* mutant is actually selfish. It is deleterious, but does it have a transmission advantage? This should be discussed here when it is first introduced.

We apologize for the confusion. We did not mention the selfish nature of the *mt:Col^{ts}* mutant at this point because the chapter focuses only on purifying selection.

Moreover, the selfish transmission advantage of the *mt:Col^{ts}* mutant only manifested when paired with more diverged functional mitochondrial genomes, which have very different non-coding regions. We therefore think that it is more appropriate that we discuss the selfish behavior of the *mt:Col^{ts}* mutant in the “selfish selection” chapter.

To make it clearer, we added a few sentences to explain why the *ts* mutant is not selfish when paired with the closely-related wild-type mtDNA in the “selfish selection”

chapter. Thank you for pointing this out.

23) 5th paragraph “In a study using *D. melanogaster* heteroplasmic for both...” does “deletion molecules” refer to organelles or genomes? If the latter, rephrase to “deletion-bearing mt genomes”.

We have rephrased to “deletion-bearing mitochondrial genomes”.

24) 6th paragraph – similarly here, it is not clear whether these mt mutations are selfish or just deleterious. Also in this paragraph, there is a lot of mention of purifying selection not taking place. Its not that purifying selection is absent (if the mutations are deleterious than by definition they are evolving under purifying selection), it just might not be strong/efficient enough to eliminate them.

We agree with the reviewer and have rephrased to “purifying selection was not detected”.

Selfish selection

25) 2nd paragraph first/second sentence – this is another sentence in which the muddled definition of “selfish” comes through. By definition a mt genome must be deleterious to organismal fitness to be selfish.

As mentioned earlier, we disagree with the reviewer’s definition of selfish and so have maintained our phrasing.

26) 3rd paragraph “A recent study suggested that uaDf5...” Again, mention here that it appears this mutation “hijacks” the replication control mechanisms and proliferates because it reduces OXPHOS function.

We have added such a statement as suggested.

27) 4th paragraph – I think a key takeaway here is whether a mt genome has a replication advantage or not depends on what other mt genomes its competing against. Therefore, while the mt:COI mutation may be deleterious in any context, it only has a replication advantage in certain situations. The recombination result is very interesting and also suggests that the replication advantage mutation and the deleterious mutation are not necessarily the same in all cases, but may be effectively linked due to the non-recombination of the mt genome. Make this clear – that the “selfish” quality of an mt genome may be due to multiple linked variants.

We apologise for not making this clear. We have added additional descriptions to make these points more explicit.

28) 5th paragraph – is a takeaway here that nuclear background also matters as to whether a mt genome has a replication advantage or is deleterious? “Interestingly, mtDNA from *D. mauritiana*...” in this sentence it is not clear that the foreign mtDNA is selfish, does it have any deleterious effects?

Here the takeaway is that genomes can have selfish drive when competing against diverged genomes even from different species – 1) *D. melanogaster* mtDNA mutants outcompete functional *D. yakuba* mtDNA and 2) *D. mauritiana* mtDNA outcompeted

D. melanogaster mtDNA. Based on the published literature, *D. mauritiana* mtDNA showed no deleterious effects and we have adjusted the sentence to make this clear.

29) 6th paragraph – as mentioned above, the first sentence muddies the definition of selfish – non-deleterious mt genomes cannot be selfish.

Again, we would like to emphasize that our definition of selfish is transmission advantage independent of the OXPPOS function.

The interplay of different selections at multiple levels

30) 1st paragraph – here and throughout, consider whether the terms “purifying” and “selfish” selection might be better replaced with “organismal selection” and “selection for replication” or something similar that indicates the difference in levels of selection.

We have decided not to change the terms because we specify when we are referring to a specific level of selection and otherwise refer to selection in general and at many levels.

31) 1st paragraph – paternal leakage is quite common in some plant lineages, and has even been observed at high levels in *Drosophila* (e.g., Birky 1995 PNAS; James and Ballard 2003 Genetics)

We have excluded the additional information on plants as we have taken the reviewer’s advice and explicitly focused on animals. The two papers suggested by the reviewer are not related to paternal leakage, and we believe that we have now cited sufficient evidence for paternal leakage in the revised version.

32) 2nd paragraph – “This can happen if the selfish drive only...” does “small” in this sentence imply that if the variant gives a very strong replication advantage they will never reach high levels? It is a bit confusing.

We have rephrased the sentence to make it more explicit.

33) 2nd paragraph – “Selfish selection mainly occurs at the genome level, whereas...” it is unclear to me why selfish conflict could not occur at the organelle level as well as the genome level. Couldn’t a mt variant cause mitochondrial organelle proliferation, but result in less fit organelles as well? Possibly by manipulating membrane or fission cycles? Is there evidence for this?

First of all, we used “mainly” as replicative advantage of a mitochondrial genome would give it a transmission advantage, so we did not exclude other possibilities. No evidence of other type of selfish genome have been demonstrated, as far as we are aware of. All the proteins required for mitochondrial biogenesis, fusion and fission are encoded by the nuclear genome, so mtDNA has little power in manipulating these activities. However, we can’t exclude possibilities where a selfish detrimental genome may indirectly lead to an increased total mitochondrial mass to compensate for the reduced energy production.

34) Fig 3. – at the top right, I suggest adding a comparative panel with a male beneficial mutation to show that because males do not transmit mt genomes, both mt

genomes will reach the same level in the next generation regardless of their effects on males.

Here, our main focus is to illustrate how detrimental mutations can reach high levels in a given population, so we have decided to leave out the additional panel to keep the figure as simple as possible. However, we have added a sentence in the main text stating that males are dead-end for mtDNA transmission.

35) 3rd paragraph beginning with “The nuclear background can influence the strength of purifying...” The entire point of this paragraph was unclear to me. It is unclear what is meant by “levels of proteins” “nuclear isoforms” or “this would reduce the need and strength of purifying selection”. There may be important points in this paragraph, but it was largely confusing to me what the authors were trying to get across.

We have made some changes to make it more explicit.

Heteroplasmy and mitochondrial replacement therapy

36) 1st paragraph – I think there are actually a handful of three-parent babies that are now teenagers. These procedures were done illegally I believe and are not well-documented, but I think the authors should look into this and make this sentence reflect the history or MRT, although I’m sure it’s a bit murky. See this popular article: <https://slate.com/technology/2016/02/three-parent-babies-have-been-here-since-the-late-90s.html>

The three-parent babies the reviewers mentioned were generated by cytoplasmic transfer, rather than nuclear transfer as described in our review. However, it is a very relevant point to be included as these people are also heteroplasmic for their mtDNA, probably with even higher level of maternal mtDNA. We have included a discussion to include those cases in both 2nd and 3rd paragraphs.

37) Last paragraph – “If we can make sure that the...” the idea of selecting a donor with an mt genome that will have a replication advantage is really interesting, but you might want to reiterate here that mitonuclear interactions might want to also be considered in selecting a donor. For example, an ideal donor would have a mt genome that is compatible with the patient’s nuclear DNA and has a replication advantage against its defective mtDNA.

We have added such a discussion.

38) One thing that isn’t clear in this last section is why does carryover occur at all? Is this just because the laboratory techniques are imperfect and a bit of the patient’s cytoplasm with some mitochondria is always taken by accident? Would there be a way to eradicate the patient’s cytoplasm/mt genomes prior to adding the donor cytoplasm?

Although we are not expert in nuclear transfer, we believe that the carryover is mainly due to the imperfect lab techniques. Moreover, some mitochondrial populations might be physically adjacent to the oocyte nucleus. It is therefore difficult to avoid maternal mitochondria completely during the transfer even with the steadiest hands.

Referee: 2

Comments to the Author(s)

Title:

A battle for transmission: the cooperative and selfish mitochondrial genome

Authors:

Anna Klucnika and Hansong Ma

Summary:

In this manuscript, Klucnika and Ma review selfish behavior of mitochondrial genomes in bilaterian animals, with especial focus on human mitochondrial dynamics. The review discusses a variety of related topics, including heteroplasmy, selection, drift, and bottlenecking in the oocyte. Notably, the evidence described, while thoroughly researched, is restricted to bilaterian animals. This choice has the unintended effects of both limiting the topic to a relatively narrow section of mitochondrial diversity, and trying to pigeonhole mitochondrial biology of nematodes and insects onto humans.

Overall, I found the topic to be of broad interest to the readership of Open Biology, and if it is recast and re-titled to focus more narrowly on heteroplasmy in bilaterians, to be quite well done. I also think the manuscript could use more careful editing, as there were several language usages that I found odd and unclear. Personally, I find the focus on MRT to be less interesting than the dynamics of heteroplasmy, but the authors do provide a novel perspective on it. Assuming the authors can narrow the scope of their review and address a couple other minor issues, I think this manuscript would make for a nice contribution to Open Biology.

Major Concerns:

1. Framing of the manuscript. The manuscript in its current state is framed as a broad look into mitochondrial biology; however, little if any of the references discuss mitochondrial biology outside of mammals, nematodes, and insects. This choice to ignore the vast majority of mitochondrial biology while trying to draw broad conclusions is odd. Much of what the authors are trying to discuss has been researched extensively in other taxa, especially plants, but no credit is given to the work done in that field. For example, mitochondrial genomes often experience rampant recombination, varying degrees of biparental inheritance exist in non-bilaterian lineages, and mutation rate varies orders of magnitude across taxonomic groups, even within metazoa. This, as one might imagine, has profound consequences for mitochondrial biology, especially with respect to the presumed mutational meltdown mitochondrial genomes are supposed to experience. Accordingly, the authors need to either broaden their literature scope considerably or restrict their observations to bilaterian animals. I have also included a couple citations below whose inclusion would help improve the manuscript, even if the focus becomes narrower.

We totally agree with the reviewer about ignoring the vast majority of mitochondrial biology literatures, especially the plant work. We intended to just describe mtDNA heteroplasmy dynamics in animals and used the word metazoa in the abstract, but failed to make it clear for majority of the manuscript. We apologize for the mistake and providing a misleading broad impression. Therefore, we have changed the title to: A battle for transmission: the cooperative and selfish animal mitochondrial genomes. We have also made changes throughout the manuscript to restrict our

discussions on bilaterian animals, and cited some of the papers recommended by the reviewers at the right places.

2. Emphasis on purifying selection/genetic drift. The section describing the forces that can affect mitochondrial genome evolution and selfish mitochondrial genome proliferation was overly simplistic. While purifying selection can and does play a major role in mitochondrial genome evolution, the authors neglect entirely the role of positive selection in mitochondrial biology. A poignant example can be glimpsed in mitochondria present in populations that have migrated to low-oxygen conditions (e.g., high elevation) relative to their ancestors and/or close relatives. While the mitochondrial genome is small and rates of recombination are low, beneficial mutations can still experience positive selection, especially when mitochondria are poorly adapted to local environmental conditions (i.e., when fitness effects of mutations are large).

This is a very good point. We have rephrased the title of the purifying selection section to: Selections for organismal fitness, so we can talk about both purifying and positive selections. We have also devoted a paragraph in this section to discuss positive selection.

While much of the manuscript was well done, I have a couple additional minor comments that the authors might consider to improve their manuscript.

- Page 2, when describing the genetic content of mitochondrial genomes. Note that this section applies exclusively to bilaterians. Mitochondrial genomes of non-bilaterians can differ quite dramatically from those of the systems that are the focus of the present study (e.g., SDH is commonly still mitochondrially encoded in many plant lineages)

As mentioned earlier, we have restricted our discussion on bilaterian mtDNA and have made this explicit in the revised manuscript. We have now also cited the 2015 Smith & Keeling PNAS review that summarises the diversity of mtDNA organization and gene content.

- Similarly, the point about high mutation rates does not apply beyond bilaterians. See Lavrov and Pett for a nice review on metazoan mitochondrial genome diversity.

We have rephrased “high mutation” to “constantly-occurring mutations”.

- First paragraph of page 3 – “so the genome is under the strong influence of genetic drift”. I think a lot of mitochondrial researchers would object strongly to this, absent some contextual justification. See Konrad et al. 2017 for a contradictory opinion.

There are numerous papers supporting the drift ideas (e.g. Wonnapijit et al, 2008, Am J Hum Genet; Brown et al, 2001 Am J Hum Genet; Jenuth et al, 1995 Nat Genet). Therefore, we would like to point out the influence of genetic drift on heteroplasmic dynamics. However, to ensure that we are inclusive of different ideas in the field, we have changed the statement to “so the genome can be under the strong influence of genetic drift”, and cited papers with contradictory opinions including the Konrad et al. paper.

- There is a repeated use of the term “selections”. I believe it would be better to say “types of selection”.

We agree with the reviewer, and have made the changes throughout the manuscript when appropriate.

- Page 4 “The nuclear genome is thus interested in...”. This is another problematic sentence formulation. The nuclear genome is not interested in anything. Nuclear genomes that successfully coordinate mitochondrial expression/replication/transmission have higher fitness than nuclear genomes that do not.

We have deleted the sentence.

- Cite Holland et al. 2018 on page 4 somewhere.

We have added the reference.

- “Limited recombination has little power to remove de novo mutations”. I do not believe this statement to be factually correct. See Neiman and Taylor 2009. Only a little bit of recombination is necessary to reset Muller’s Ratchet.

We have rephrased the statement to “uniparental inheritance and little recombination has limited power of removing *de novo* mutations”.

- I think the phrase “selfish selection” is odd. I would change to “Selfish proliferation”. While selection can act at the level of the replicator (i.e., rapid/efficiently replicating mtDNA molecules have a fitness advantage over slowly replicating mtDNA molecules), I think the phrasing here sparks of some active role of the molecule in its own selection. The term “proliferation” better encapsulates the phenomenon.

We appreciate the reviewer’s suggestion. However, phrases like “selfish proliferation” imply that the nature of selfish transmission advantage is mainly due to replicative advantage. There might be other mechanisms that lead to selfish behaviors that we do not want to exclude. Furthermore, there are many examples where selfish elements have evolved active mechanisms to increase their abundance by interference or gonotaxis (summarised in *Genes in Conflict* book) (i.e. active role of the DNA molecule in its own selection). For example, the killer X chromosomes can increase in frequency by interfering with the replication of the alternative allele. Some B chromosomes show preferential movement to the germline. We therefore argue that the phrase selfish selection is appropriate, especially when describing evolution in a gene-centric view. However, we have defined selfish selection when we first mentioned it as “selection for selfish gains in transmission”.

- The lack of any information about cytoplasmic male sterility in plants seems a glaring omission in the selfish element section. This is a major reason why I think the manuscript needs to be re-framed/narrowed in scope.

We again apologise for not making it clear that our review focus on only bilaterian mtDNA. We have made it clear in the revised version.

- Remove the reference to unpublished data from the top part of page 9. Given that it is unpublished, the audience cannot properly evaluate the quality of the evidence supporting the point. The resulting conclusion should therefore be removed until the data can be properly viewed.

We agree and have removed the statement that is supported by unpublished data.

- The description of mitonuclear incompatibility and subsequent compensatory coevolution is not correctly described. See Sloan et al. 2017 for a better description of the dynamics at play.

We have taken on board the reviewer's comments and have rephrased our description. We hope that the description is now correct and that we have made the focus on heteroplasmy more explicit.

- In the MRT section, I think it would be worth citing Havird et al 2016.

We have cited the paper.

Recommended literature:

Barr et al.. 2005. Inheritance and recombination of mitochondrial genomes in plants, fungi and animals. *New Phytol.* 168:39-50.

Havird et al., 2016. Sex, mitochondria, and genetic rescue. *TREE* 31: 96-99.

Holland et al., 2018. Deep-Coverage MPS Analysis of Heteroplasmic Variants within the mtGenome Allows for Frequent Differentiation of Maternal Relatives. *Genes.* 9: 124.

Konrad et al., 2017. Mitochondrial Mutation Rate, Spectrum and Heteroplasmy in *Caenorhabditis elegans* Spontaneous Mutation Accumulation Lines of Differing Population Size. *MBE.* 34: 1319–1334.

Lavrov and Pett 2016. *GBE.* Animal mitochondrial DNA as we do not know it: mt-genome organization and evolution in nonbilaterian lineages. 8: 2896-2913.

Neiman and Taylor 2009. The causes of mutation accumulation in mitochondrial genomes. *Proc Roy Soc B.* 276: 1201-1209.

Sloan et al., 2017. The on-again off-again relationship between mitochondria and species boundaries.